# CogniPair: GNWT-inspired cognitive architecture for generative agents for Social Pairing - Dating & Hiring Applications

## Abstract

Current large language model agents lack authentic human psychological processes necessary for genuine digital twins. We present the first computational implementation of Global Workspace Theory (GNWT), creating agents with multiple specialized sub-agents (emotion, memory, social norms, planning, goal-tracking) coordinated through a global workspace broadcast mechanism. This architecture allows agents to maintain consistent personalities while evolving through social interaction. Our CogniPair simulation platform deploys 551 GNWT-Agents for speed dating interactions, grounded in real data from the Columbia University Speed Dating dataset. Evaluations show strong psychological realism, with agents achieving 72% correlation with human attraction patterns and outperforming baselines in partner preference evolution (72.5% vs. 61.3%). Human validation studies confirm our approach's fidelity, with participants rating their digital twins' behavioral accuracy at 5.6/7.0 and agreeing with their choices 74% of the time. This work establishes new benchmarks for psychological authenticity in AI systems and provides a foundation for developing truly human-like digital agents.

## 1 Introduction

Human social interactions—from dating to job interviews—require not just coherent dialogue but authentic psychological processes including emotion regulation, memory consolidation, and dynamic preference formation. LLM-based agents have been applied to model human social interactions, showing promise in domains such as customer service, healthcare assistance, and educational tutoring Park et al. (2023); Xu et al. (2023); Wang et al. (2024). Despite recent advances, current LLM-based agents face two fundamental limitations that restrict their ability to model human behavior realistically: (1) the **psychological behavior gap**—they cannot authentically simulate internal mental states, emotional processing, or evolving preferences Zhang et al. (2024); Serapio-García et al. (2023); Jiang et al. (2023); Guo et al. (2024); and (2) the **social behavior gap**—they fail to capture the complex dynamics of human-to-human interactions where preferences and behaviors co-evolve through social experiences Park et al. (2023); Zhang et al. (2018); Huang et al. (2024); Park et al. (2022); Aher et al. (2023); Sun et al. (2024).

The **psychological behavior gap** manifests in two critical problems: the **individualization problem**, where agents act like generic humans rather than specific individuals with unique psychological profiles, and the **static personality problem**, where agents cannot evolve mentally through experience. Existing approaches such as Stanford's Generative Agents Park et al. (2023) demonstrated emergent behaviors but relied on fictional personas without real human data. PersonaChat Zhang et al. (2018) introduced personality descriptions that remain synthetic and fixed. Recent personality modeling efforts Serapio-García et al. (2023); Jiang et al. (2023) achieve only surface-level behavioral mimicry without cognitive grounding. Most critically, these approaches treat personality as immutable prompts rather than dynamic psychological states shaped by experience.

The **social behavior gap** emerges when attempting to model authentic social interactions, particularly in complex domains such as relationship formation. Current LLM agents lack the capability to engage in authentic social dynamics where preferences evolve through interaction, emotional responses adapt to social feedback, and behavioral patterns shift based on interpersonal experiences. This limitation

becomes particularly pronounced in domains requiring complex social cognition, such as dating scenarios where mutual attraction emerges through dynamic, bidirectional assessment processes.

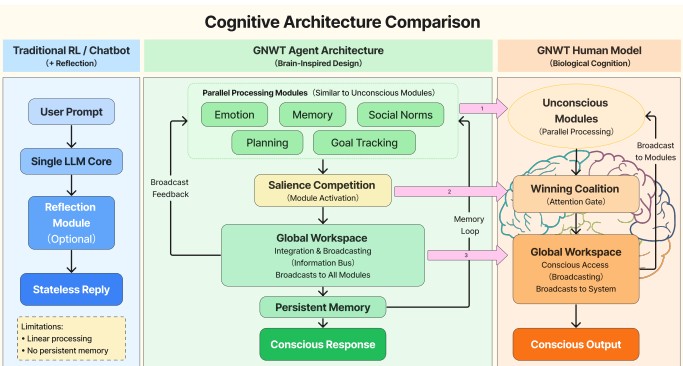

Figure 1: Comparison of Cognitive Architectures: Traditional RL/Chatbot (left), GNWT Agent Architecture (center), and GNWT Human Model (right), showing the evolution from linear processing to brain-inspired parallel processing with global workspace integration.

To address these fundamental gaps, we turn to Global Workspace Theory (GNWT) Mashour et al. (2020); Baars (2005), a leading neurocognitive model that explains how human consciousness emerges from the interaction of specialized brain modules. In human cognition, GNWT describes how disparate neural processes—emotion, memory, perception, and planning—compete for access to a central "global workspace." When information becomes sufficiently salient, it triggers a broadcast that propagates this content throughout the brain, creating our unified stream of consciousness. This theoretical foundation provides a clear roadmap for building agents that can overcome both the psychological and social behavior gaps(Figure 1). To address the **psychological behavior gap**, we operationalize GNWT into a computational agent architecture where each individual agent contains multiple specialized sub-agents working in parallel as a unified consciousness. Our GNWT-Agent implements five specialized cognitive modules—Emotion, Memory, Planning, SocialNorms, and GoalTracking—each grounded in neurocognitive theories and parameterized by the agent's Five-Factor personality profile. By implementing GNWT's broadcast mechanism computationally, we create agents with genuine internal psychological dynamics—emotion sub-agents generate affective responses, memory sub-agents consolidate experiences, social norms sub-agents manage cultural awareness, planning sub-agents develop strategies, and goal-tracking sub-agents maintain objectives. This architecture fundamentally differs from traditional LLM agents that process inputs sequentially without internal state evolution.

To bridge the **social behavior gap**, we developed **CogniPair**, a social-influence decision system that enables GNWT-Agents to engage in authentic social interactions and evolve through experience. CogniPair is not merely a testbed but a comprehensive system for modeling and guiding social influence between individuals, ultimately optimizing decision processes across various social contexts. While our primary evaluation uses a speed dating testbed, the CogniPair system itself can be extended to other social decision environments such as team formation, negotiation scenarios, and collaborative problem-solving.

We selected speed dating as our evaluation domain because it exemplifies the most challenging aspects of human social cognition—rapid compatibility assessment, dynamic preference formation, emotional regulation under uncertainty, and integration of multiple information streams. The Columbia University Speed Dating dataset Fisman et al. (2006; 2008) provides rich behavioral ground truth, including pre- and post-interaction preferences, attraction ratings, and decision outcomes, enabling rigorous evaluation of social realism. Our dating testbed deploys 551 GNWT-Agents in a two-level simulation architecture: the internal level models psychological processes within each agent, while the external level simulates social dynamics between agents. This scale—20 times larger than previous personality-based simulations Park et al. (2023)—enables statistically valid analysis of emergent social patterns, creates sufficient diversity for complex relationship networks, and allows measurement of population-level phenomena.

Through this dual approach—addressing the psychological gap with GNWT-Agents and the social gap with the CogniPair system—our framework uniquely enables both individualization and dynamic evolution. The global workspace mechanism naturally handles the stability-plasticity dilemma in personality modeling: core traits remain stable through persistent attention patterns while allowing adaptive changes through experience-driven broadcast priority shifts. Unlike previous approaches that treat personality as static prompts or surface behaviors, our architecture models the cognitive processes underlying personality and the social dynamics shaping its evolution.

Our evaluation framework measures psychological and social realism across multiple dimensions. For psychological realism, we assess preference consistency (how well agents maintain core values while adapting), emotional coherence (whether affective responses follow human psychological principles), and memory integration (how past experiences influence decisions). For social realism, we measure attraction correlation (how closely agent mate selection aligns with human patterns), interaction dynamics (how conversations evolve), and emergent social phenomena (group formation, preference shifts). These metrics ground our claims about achieving human-level authenticity in both psychological processing and social behavior.

Our experiments demonstrate that GNWT-Agents closely capture human social dynamics, showing improvements across multiple evaluation dimensions. We show significant improvements in partner preference evolution (72.5% accuracy vs. 61.3% for Multi-Agent Debate), self-perception adaptation, external evaluation changes, and match prediction (77.8% accuracy) compared to state-of-the-art baselines. Our agents demonstrate human-like evolution patterns with high correlation to ground truth data (above 0.7 across multiple dimensions, with a 0.72 correlation for match patterns). Human validation studies further confirm the psychological fidelity of our approach, with participants rating their digital twins' behavioral fidelity at 5.6/7.0 and agreeing with their twin's choices 74% of the time. Our key contributions are:

1. We are the **first to operationalize GNWT for computational agents**, creating a cognitive architecture where multiple sub-agents within each agent replicate human psychological processes through dynamic workspace broadcasting

2. We develop CogniPair, the **first social-influence decision system combining cognitive theory with large-scale social simulation**, capable of generalizing beyond our dating testbed to various social decision environments as demonstrated by successful transfer to job interview contexts (81% accuracy)

3. Our extensive experiments confirm GNWT's broadcast mechanism enables **genuine personality evolution**, with significant improvements in partner preference evolution (72.5% vs. 61.3%), self-perception adaptation, and external evaluation shifts compared to state-of-the-art baselines

## 2 RELATED WORK

**LLMs for Social Simulation:** Recent advances in LLMs have enabled sophisticated conversational agents Thoppilan et al. (2022); Park et al. (2023), yet standard models lack persistent psychological states Guo et al. (2024); Zhang et al. (2024) and show limitations in capturing social dynamics Xi et al. (2023); Wang et al. (2024). Reasoning enhancements such as Chain-of-Thought Wei et al. (2022), self-consistency Wang et al. (2023a), retrieval-augmentation Lewis et al. (2020), and memory architectures Guo et al. (2024); Zhang et al. (2024) improve performance but rarely incorporate selective attention Zhang et al. (2024); Zhu et al. (2023); Andreas (2022). Social simulation platforms (e.g., Generative Agents Park et al. (2023), PersonaChat Zhang et al. (2018), Li et al. Li et al. (2024), Gao et al. Gao et al. (2023)) implement memory or persona-based interaction, but typically rely on fictional characters without psychological grounding. Empirical studies Huang et al. (2024) highlight gaps between simulated and human behavior, and recent evaluations Aher et al. (2023); Park et al. (2023) note the lack of real data, neuroscience-based cognition, adaptive personality evolution, or comprehensive social metrics. Our CogniPair system integrates all these missing components.

**Modeling Psychological Processes:** Traditional cognitive architectures relied on symbolic rules, while Global Neuronal Workspace Theory (GNWT) Baars (2005); Mashour et al. (2020) offers a framework for dynamic attention. Computational implementations Bengio (2017); Goyal et al. (2022); Dehaene et al. (2017); Mashour et al. (2020) focus mainly on perception rather than higher-

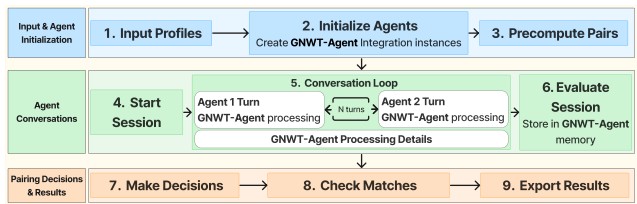

Figure 2: CogniPair Platform System Flow

order social cognition. Digital twin research emphasizes behavioral mimicry Park et al. (2024), and personality modeling systems Li et al. (2023); Wang et al. (2023b); Acerbi & Stubbersfield (2023); Sun et al. (2024) often treat traits as static. Multi-agent approaches using debate Du et al. (2023); Chan et al. (2023) or transformer-based aggregation Chen et al. (2023); Hong et al. (2023) improve over single-agent methods but rely on explicit turn-taking rather than parallel selective attention. Our GNWT-Agent instead leverages a global workspace for personality-driven prioritization, while CogniPair enables realistic preference adaptation and contextual coherence, evaluated with the Columbia Speed Dating dataset Fisman et al. (2006).

## 3 CogniPair: Cognitive Social Pairing Agent System

In this section, we introduce single GWNT-Agent's structure (Sec. 3.1), how the single-turn conversation generated from GNWT-Agent (Sec. 3.2), and simulated social environment that allows multi-turn conversations (Sec. 3.3),

### 3.1 GNWT-Agent Cognitive Modules

**Core Cognitive Architecture:** GNWT-Agent's cognitive processing is based on the Global Neuronal Workspace Theory (GNWT), which provides a computational model of human consciousness and cognitive processing mechanisms (Algorithm 1, Figure 6). In implementation, we deploy five competing cognitive modules, each focusing on different cognitive functional domains(detailed system flow in Appendix A.3)):

**Emotion Module**. This module performs in three stages: (1) emotion detection, identifying affective markers in text through feature extraction; (2) valence-arousal assessment, mapping detected emotions to a two-dimensional affective space; and (3) regulation strategy generation, adjusting emotional response intensity based on the agent's neuroticism ($N$) parameter. Mathematically represented as: $R_{\text{Emotion}} = f_{\text{E}}(Q, H, \text{GW}, N)$, where higher $N$ values amplify emotional processing weights.

**Memory Module**. This module maintains a dual memory system: (1) episodic memory, storing time-stamped dialogue segments and interaction patterns; and (2) semantic memory, preserving knowledge about conversation topics and abstract concepts. The retrieval process employs vector similarity search, with the openness parameter ($O$) adjusting memory breadth and retrieval strategies: $R_{\text{Memory}} = f_{\text{M}}(Q, H, \text{GW}, O)$.

**Planning Module**. This module implements hierarchical goal decomposition: breaking down complex social goals (e.g., "establish rapport") into tactical steps (e.g., "identify common interests," "express empathy"). This process is regulated by the conscientiousness parameter ($C$), which controls planning depth and strategic rigor: $R_{\text{Planning}} = f_{\text{P}}(Q, H, \text{GW}, C)$.

**SocialNorms Module**. This module maintains a knowledge base of social interaction rules, evaluating the appropriateness of conversational behaviors. Processing includes: (1) etiquette checking, verifying response politeness; (2) boundary monitoring, preventing excessive self-disclosure; and (3) reciprocity verification, ensuring balanced conversational contributions. The agreeableness parameter ($A$) adjusts the strictness of norm enforcement: $R_{\text{SocialNorms}} = f_{\text{SN}}(Q, H, \text{GW}, A)$.

**GoalTracking Module**. This module continuously evaluates conversational progress: (1) direction monitoring, tracking advancement toward preset objectives; (2) uncertainty assessment, identifying information gaps that require clarification; and (3) direction adjustment, recalibrating goals based on interaction dynamics. The extraversion parameter ($E$) influences goal assertiveness: $R_{\text{GoalTracking}} = f_{\text{GT}}(Q, H, \text{GW}, E)$.

## 3.2 SINGLE DECISION-MAKING SYSTEM FLOW

The CogniPair system implements a structured decision-making flow that systematically processes social interactions through nine distinct stages across three operational phases, as illustrated in Figure 2 & Algorithm 2.

**Phase 1: Input & Agent Initialization** establishes the foundation for social simulations. In Step 1 (*Input Profiles*), the system ingests personality profiles, preference distributions, and demographic data, either from human participant records (e.g., the Columbia Speed Dating dataset) or synthetically generated profiles with balanced demographic distributions. Step 2 (*Initialize Agents*) instantiates GNWT-Agent cognitive architectures for each participant, mapping Five-Factor personality traits to module weights and initializing the global workspace with prior knowledge. This process follows $\text{Agent}_i = \text{INITIALIZEAGENT}(\text{Profile}_i, \theta_{\text{module}})$, where $\theta_{\text{module}}$ represents the module-specific parameters. In Step 3 (*Precompute Pairs*), the system generates potential interaction dyads based on specified criteria (e.g., gender preferences, age constraints), creating a pairing matrix $P_{m \times n}$ where each element $p_{ij}$ indicates pairing eligibility between agents $i$ and $j$.

**Phase 2: Agent Conversations** executes the multi-turn interactive dialogues. Step 4 (*Start Session*) initializes the conversational context $C_0$ with environmental parameters (e.g., spatial configuration, temporal constraints) and interaction goals. Step 5 (*Conversation Loop*) implements the turn-taking dynamics where each agent processes inputs through their cognitive modules and generates responses. For each turn $t$, Agent 1 generates response $R_{1,t} = \text{PROCESSINPUT}(Q_t, H_{t-1}, \text{GW}_1, P_1)$ where $Q_t$ is the current query, $H_{t-1}$ is the conversation history, $\text{GW}_1$ is Agent 1's global workspace state, and $P_1$ represents personality parameters.

Within each processing cycle, the agent integrates outputs from all cognitive modules to form a coherent response through the INTEGRATEMODULEOUTPUTS function (Algorithm 1 Line 29), formalized as a personality-weighted combination:

$$\text{Response} = \text{INTEGRATEMODULEOUTPUTS}(\{R_M\}, \text{GW}, P) \tag{1}$$

$$= \sum_{M \in \text{Modules}} \alpha_M(P) \cdot R_M + \beta(P) \cdot G(\text{GW}) \tag{2}$$

where $\alpha_M(P)$ represents personality-based module weights, $\beta(P)$ is the integration coefficient for global workspace content, and $G(\text{GW})$ extracts key content from the global workspace. This combination strategy ensures the final response reflects both specialized processing from each module and maintains global coherence.

Agent 2 follows an identical process, creating a bidirectional exchange repeated for $N$ turns. Each GNWT-Agent processing instance involves all five cognitive modules competing for global workspace access, with broadcasts occurring when salience exceeds the threshold $\tau$. During processing, emotional reactions, memory retrieval, planning strategies, social norm evaluations, and goal assessments are computed in parallel, with integration weighted by personality parameters.

In Step 6 (*Evaluate Session*), agents assess the interaction quality through multiple dimensions: $E_i = \{E_{\text{attr}}, E_{\text{similar}}, E_{\text{comfort}}, E_{\text{interest}}\}$, with these evaluations stored in the Memory module for subsequent retrieval. The system's adaptive learning mechanism is implemented through two key update functions: (1) the Memory.UPDATELONGTERM($Q$, Response, $H$) function (Algorithm 1 Line 30) stores the current interaction in long-term memory, using attention-based memory consolidation techniques that highlight emotionally salient and goal-relevant content; and (2) the UPDATEPREFERENCES($P$, $Q$, Response, $H$) function (Algorithm 1 Line 31) adjusts personality weights based on interaction experiences, implementing fine-tuning learning:

$$P_{t+1} = P_t + \eta \cdot \nabla_P J(P_t, Q, \text{Response}, H) \tag{3}$$

where $\eta$ is a learning rate parameter and $\nabla_P J$ is the gradient of an objective function measuring interaction success with respect to personality parameters. This dual update mechanism enables agents to continuously evolve their preferences and behaviors based on accumulated social experiences.

**Phase 3: Pairing Decisions & Results** culminates in match determinations. In Step 7 (*Make Decisions*), each agent formulates a binary decision (accept/reject) regarding potential future interactions: $D_i = \text{DECISIONFUNCTION}(E_i, P_i, H, \text{GW}_i)$, where the decision function integrates evaluation metrics, personality preferences, and interaction history. Step 8 (*Check Matches*) identifies

mutual matches where both agents express interest: $M_{ij} = D_i \wedge D_j$, creating a symmetric match matrix. Finally, Step 9 (*Export Results*) aggregates and formats simulation outcomes, including match decisions, preference evolutions, perception changes, and interaction quality metrics, generating comprehensive datasets for subsequent analysis.

This workflow implements the three complexity levels described earlier: low-complexity interactions utilize direct module selection, moderate-complexity interactions employ iterative processing with conflict resolution, and high-complexity interactions integrate the complete multi-phase protocol with comprehensive state tracking. CogniPair's decision-making and learning mechanisms are deliberately scenario-agnostic, allowing application across diverse social contexts—from optimizing information exchange in professional settings to fostering emotional connection in personal relationships. Through this systematic approach, CogniPair captures both the cognitive micromechanics of individual decision-making and the emergent macropatterns of social pairing dynamics.

### 3.3 SIMULATED SOCIAL ENVIRONMENT SETUP

#### 3.3.1 GENERALIZED ENVIRONMENT PARAMETERIZATION

CogniPair implements a flexible parameterization system for modeling diverse social interaction environments. The system encapsulates interaction contexts through a comprehensive parameter space $C$ defined as:

$$C = \{\text{physical}_p, \text{temporal}_t, \text{social}_s, \text{cultural}_c\} \tag{4}$$

$$\text{physical}_p = \{\text{spatial\_layout}, \text{proximity}, \text{sensory\_conditions}\} \tag{5}$$

$$\text{temporal}_t = \{\text{duration}, \text{pacing}, \text{sequence\_structure}\} \tag{6}$$

$$\text{social}_s = \{\text{group\_size}, \text{relationship\_dynamics}, \text{power\_structure}\} \tag{7}$$

$$\text{cultural}_c = \{\text{normative\_expectations}, \text{communication\_styles}\} \tag{8}$$

The physical parameters capture environmental conditions including spatial arrangements, interpersonal distance, and sensory factors (lighting, acoustics, temperature) that influence interaction dynamics. Temporal parameters define interaction timeframes, turn-taking pacing, and structural sequencing that shape conversational flow. Social parameters model group composition, pre-existing relationship dynamics, and authority structures. Cultural parameters encode normative behaviors and communication conventions appropriate to specific contexts.

This generalized parameterization enables CogniPair to simulate diverse interaction scenarios—from professional meetings to casual gatherings, educational exchanges to intimate conversations—by appropriately configuring these parameters. The system generates contextually-appropriate prompts using natural language templates that translate numerical parameter values into detailed environmental descriptions accessible to language models, enhancing validity across different simulation contexts.

#### 3.3.2 MULTI-AGENT INTERACTION ARCHITECTURE

CogniPair's interaction architecture (Algorithm 1) provides a flexible framework for simulating multi-agent social dynamics across diverse scenarios. The system initializes a pool $\mathcal{A}$ of agents, each equipped with five cognitive modules and a global workspace. The initialization process maps individual agent characteristics into module weights and interaction preferences:

$$\text{IA}_i = \text{InitializeInteractionAttributes}(P_i) \tag{9}$$

$$= \{w_1, w_2, ..., w_n\} \tag{10}$$

$$\text{Modules}_i.\text{weights} = M(P_i, \text{IA}_i) \tag{11}$$

where $P_i$ represents the agent's personality profile (typically Five-Factor traits), $\text{IA}_i$ represents scenario-specific interaction attributes, and $M$ is a mapping function that determines module processing parameters. This approach ensures individual agents retain consistent core traits while adapting their behavior appropriately to different social contexts.

The interaction protocol supports multiple engagement patterns, including dyadic exchanges (one-to-one interactions with reciprocal turn-taking), group discussions (multi-participant exchanges with dynamic speaker selection), and hierarchical interactions (structured exchanges with defined

role-based communication paths). In supporting these engagement patterns, the system manages turn-taking, tracks interaction histories, and computes evolving relationship metrics, while the architecture records comprehensive data such as complete interaction histories $\mathcal{H}$, cognitive trace datasets $\mathcal{T}$ capturing internal mental states, relationship development trajectories $\mathcal{R}$ tracking interpersonal dynamics, and emergent social network structures $\mathcal{N}$ documenting group formation. This multi-level data collection enables both micro-analysis of individual cognitive processes and macro-analysis of emergent social patterns, providing a foundation for validating the system's fidelity to human social behavior across different interaction contexts.

## 4 EXPERIMENTS AND RESULTS

### 4.1 EXPERIMENTAL SETUP

**Dataset and Simulation Protocol:** The Columbia University Speed Dating dataset Fisman et al. (2006) contains records of 551 participants who engaged in 5,500+ four-minute speed dates over 21 sessions, resulting in over 8,300 observations. Each record includes pre-dating attribute self-ratings (1-10 scale), attribute importance ratings (distributing 100 points across 6 attributes), post-dating partner ratings on the same attributes, and match decisions (yes/no interest in seeing a partner again). The six key attributes measured are: attractiveness, sincerity, intelligence, fun, ambition, and shared interests. We instantiate 551 GNWT-Agents as digital twins of the original participants, initializing each agent's personality profile with the Five-Factor traits inferred from participants' self-ratings and importance distributions. The physical and temporal parameters in the CogniPair system are configured to match the original study's environment (bar-restaurant setting, 4-minute interaction, 8-10 conversation turns). For each simulated date, agents engage in 8 conversation turns, after which they update their self-ratings based on interaction experience, rate their partners on the six attributes, and make match decisions. We compare match patterns, preference evolution, and conversation dynamics against ground truth data from the original study.

**Baselines:** We compare against state-of-the-art approaches: Single Sequential LLM (standard prompt-based approach without specialized cognitive modules), Memory-Enhanced LLM (incorporates retrieval-augmented context) Lewis et al. (2020), Multi-Agent Debate (simulates internal deliberation through multiple agents) Chan et al. (2023), and Hierarchical Architecture (uses a command structure to organize decision-making) Du et al. (2023). All baselines use identical data initialization to ensure fair comparison.

### 4.2 RESULTS AND KEY FINDINGS

We examine population-level social dynamics to assess how well our system bridges the social behavior gap identified in our introduction. Our experimental approach creates digital twins of the Columbia Speed Dating study participants and compares their evolution with the ground truth human data across multiple time points.

#### 4.2.1 EVOLUTION OF SOCIAL DYNAMICS

We evaluate our system's ability to model four key dimensions of social dynamics evolution: partner preference changes, self-perception adaptation, external evaluation shifts, and match decision accuracy. As illustrated in Figure 3, our analysis reveals several important patterns that characterize human social dynamics and demonstrates the remarkable similarity in evolution patterns between human participants and our GNWT-Agents.

**For partner preferences (Figure 3A)**, both humans and agents show consistent patterns in attribute importance shifts. Attractiveness importance increases substantially ($+39.0\%$ human, $+25.0\%$ agent), while intelligence ($-24.8\%$ human, $-15.2\%$ agent) and sincerity ($-16.6\%$ human, $-10.5\%$ agent) show significant decreases. Shared interests show comparable positive changes ($+9.8\%$ human, $+9.7\%$ agent), while fun remains relatively stable with slight increases ($+1.3\%$ human, $+5.8\%$ agent). Ambition ($-7.0\%$ human, $-4.5\%$ agent) exhibits moderate decreases, suggesting its relative stability as a core value less influenced by short-term interactions.

**For self-perception (Figure 3B)**, there is subtle but consistent calibration of traits through social interaction. Unlike partner preferences, self-perception shows more conservative adjustments, with

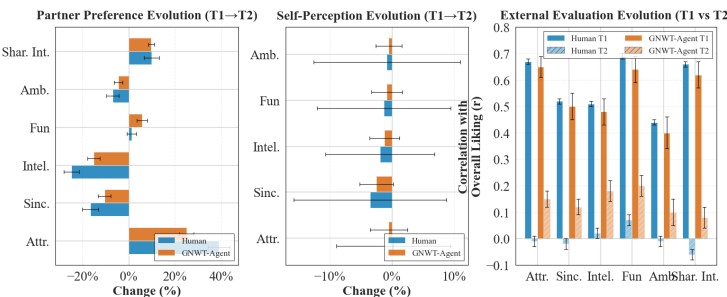

Figure 3: Social Dynamics Evolution: Human vs. GNWT-Agent Comparison. (A) Partner preference changes from T1 to T2; (B) Self-perception adjustments across attributes; (C) Evolution of attribute-liking correlations from initial to post-interaction evaluations.

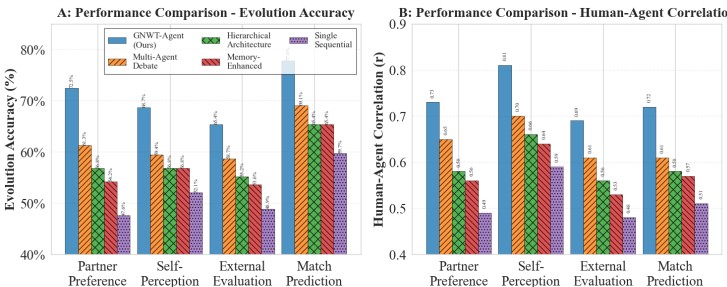

Figure 4: Comparison of GNWT-Agent with baseline methods across evolution dimensions. (A) Evolution accuracy showing GNWT-Agent's superior performance in all metrics; (B) Human-agent correlation demonstrating stronger alignment with human data compared to baseline approaches.

small negative shifts across most dimensions for both humans and agents: attractiveness ($+0.3\%$ human, $-0.5\%$ agent), sincerity ($-3.5\%$ human, $-2.5\%$ agent), intelligence ($-1.9\%$ human, $-1.2\%$ agent), fun ($-1.3\%$ human, $-0.8\%$ agent), and ambition ($-0.8\%$ human, $-0.5\%$ agent). The self-other perception gap narrows consistently for both humans and agents (from $0.8 \rightarrow 0.7$ human, $0.9 \rightarrow 0.7$ agent), reflecting the social calibration process through which external feedback helps align self-image with social reality.

**For external evaluation** (Figure 3C), there is a dramatic shift in evaluation criteria from Time 1 to Time 2. For humans, the initially strong correlations between attributes and overall liking at Time 1 (ranging from $r = 0.44$ to $r = 0.69$) diminish dramatically at Time 2 (ranging from $r = -0.06$ to $r = 0.07$), suggesting a fundamental change in evaluation criteria following interaction. Our GNWT-Agent shows a similar pattern, with high Time 1 correlations (ranging from $r = 0.40$ to $r = 0.65$) decreasing substantially at Time 2, though maintaining slight positive correlations (ranging from $r = 0.08$ to $r = 0.20$). This pattern indicates that both humans and agents undergo significant shifts in their evaluation frameworks through social interaction, though agents retain more of their initial criteria than humans do.

**Comparative Analysis:** As shown in Figure 4A, our GNWT-Agent consistently outperforms all baseline methods across evolution dimensions, with particularly strong advantages in partner preference evolution (72.5% vs. 61.3% for Multi-Agent Debate) and match prediction accuracy (77.8% vs. 69.1%). Figure 4B further demonstrates our system's superior human-agent correlation, with GNWT-Agent achieving strong correlation values above 0.7 in multiple dimensions, while baseline methods fall below this threshold. The complete comparative data tables with detailed metrics and standard deviations can be found in Appendix A.33.

### 4.3 HUMAN VALIDATION STUDIES

We validated our system by asking participants to evaluate AI versions of themselves ("digital twins") in dating and job interview contexts. Personality traits were inferred using our interactive Adventure-Based Assessment (Algorithm 3), which avoids self-report bias and correlates strongly with Big Five

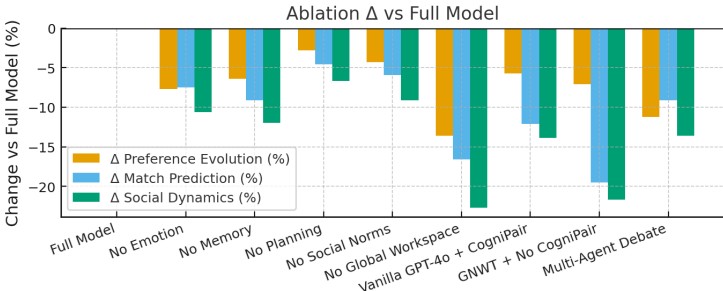

Figure 5: Ablation study: performance deltas relative to the full GNWT+CogniPair model across three metrics. Removing modules significantly reduces accuracy and correlation. Likewise, dropping the CogniPair interaction mechanism leads to large declines, confirming that both internal modules and the social platform are indispensable.

scores ($r = 0.82$, see Appendix A.32). Table 10 summarizes results: in the Speed Dating Study (20 participants), twins were rated highly realistic (5.6/7, SD=0.8) with 74% decision concordance, while the Job Interview Study (10 participants) yielded slightly higher fidelity (5.8/7, SD=0.6) and concordance (81%). Notably, twins adapted their behavior to context, showing stronger Planning influence and professional norm adherence in interviews than in dates, confirming that our framework produces recognizable digital representations that preserve personality while adjusting appropriately to different social settings.

### 4.4 ABLATION STUDIES

Figure 5 shows performance deltas relative to the full GNWT+CogniPair model. Removing the **global workspace** led to the largest drop ($-14\%$), while removing **memory** or **emotion** reduced accuracy by $-7\%$–$12\%$. Disabling social interaction was most detrimental: **GNWT without CogniPair** fell nearly $-20\%$, and a vanilla GPT-4o within CogniPair lost $-12\%$. These results confirm that *cognitive modules are complementary* and that *CogniPair is indispensable* for modeling social co-evolution.

## 5 PRIVACY PROTECTION AND LIMITATIONS

Our system faces several technical limitations, with privacy being a primary concern. To mitigate risks, we apply safeguards such as differential privacy, k-anonymization, federated learning, double-consent mechanisms, and cryptographic watermarking; extended implementations are detailed in Appendix A.34. Despite these protections, long-term concerns such as behavioral prediction and misuse remain. Other limitations include imperfect calibration of cognitive modules, limited cross-cultural robustness, the absence of non-verbal communication modeling, and scalability challenges for large populations. These primarily require refinement rather than redesign, and future work will extend cultural adaptation, incorporate non-verbal cues, and improve efficiency for broader applications in collaboration, human-AI teams, and education.

## 6 BROADER IMPACTS AND CONCLUSION

We presented the first computational implementation of Global Workspace Theory for AI agents, demonstrating psychological realism through cognitive modules coordinated by workspace broadcasting. Our GNWT-Agent closes two gaps in current AI: the *psychological gap*, by modeling authentic internal states with dynamic preference evolution, and the *social gap*, by capturing genuine interpersonal dynamics. Experiments with 551 agents showed strong alignment with humans, achieving 72% correlation with attraction patterns and 77.8% match prediction accuracy, while human validation confirmed fidelity (5.6/7.0 realism, 74% decision concordance). Beyond these results, CogniPair enables psychologically grounded matching, improved hiring fit, and richer human-AI collaboration. These contributions lay a foundation for digital twins that preserve consistent personality while evolving through experience, advancing AI systems with genuine psychological processes.

ACKNOWLEDGMENTS

We thank the anonymous reviewers for the time and effort they dedicated to reading our submission and for providing thoughtful feedback. Their comments and suggestions have helped us to clarify our presentation, strengthen our arguments, and improve the overall quality of this work. We greatly appreciate their careful engagement and constructive input throughout the review process.

ETHICS STATEMENT

This work relies on the Columbia University Speed Dating dataset (Fisman et al., 2006; 2008), which is publicly available and de-identified. All analyses were conducted with strict respect for participant privacy, and no attempt at re-identification was made. To further protect sensitive information, the CogniPair framework integrates differential privacy into personality initialization, k-anonymization of conversational traces, and federated learning to avoid central storage of individual-level data.

We recognize potential ethical concerns in generating psychologically realistic digital twins, including risks of impersonation, misattribution, and misuse. To mitigate these, all generated content is cryptographically watermarked and bounded by consent mechanisms within the system design. Broader risks such as reinforcing bias or exclusion were considered, and future work will extend fairness audits to diverse demographic settings.

In line with the ICLR Code of Ethics, our position is that the societal benefits of trustworthy, explainable social simulation outweigh these risks, provided that such safeguards and ongoing monitoring are maintained.

REPRODUCIBILITY STATEMENT

To support reproducibility, we provide detailed descriptions of the GNWT-Agent architecture (Sec. 3.1), system flow (Sec. 3.2), and simulation setup (Sec. 3.3). Algorithms and pseudocode are included in Appendix A.2, and additional worked examples and environment parameterizations are provided in Appendix A.3–A.6. All hyperparameters (e.g., learning rate $\eta$, salience thresholds $\tau$) are specified in the appendices, and baseline comparisons follow identical initialization for fairness. A complete record of evaluation metrics is presented in Appendix A.8. To further enable replication, all materials associated with this work will be made publicly available.

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

# A  APPENDIX

## USE OF LLMS

This work is centered on the use of large language models (LLMs) as core agents within the CogniPair framework. Specifically, we employed [GPT-4/other models, anonymized for review] to instantiate GNWT-based digital twins, simulate conversations, and support social pairing experiments. The LLMs were accessed through standard APIs and were not fine-tuned on private or sensitive data. Their role is not limited to auxiliary writing or coding assistance but constitutes the main experimental system under study.

In line with ICLR 2026 policy on LLM usage, we note that all analyses, design choices, and interpretations of results are the responsibility of the authors. The LLMs may introduce biases or limitations inherent to their training data, and these factors have been considered when analyzing outcomes.

## A.1 AGENT ARCHITECTURE

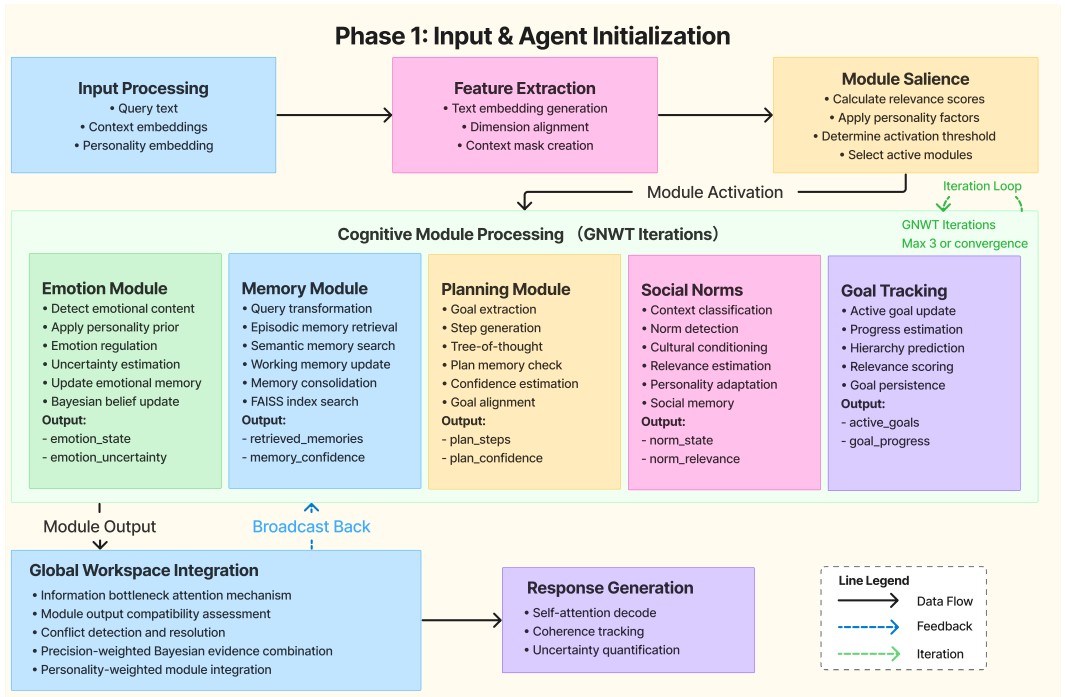

Figure 6: GNWT-Agent Architecture Internal Processing Flow

### A.1.1 LLM DETAILS

We used the OpenAI API for GPT-4o with top_p set to 1, max_tokens set to 200, min_tokens set to 0, and temperature set to 0.9 (with all other parameters at their default values),

## A.2 ALGORITHMS AND PSEUDOCODE

---

**Algorithm 1** GNWT-Agent: GNWT-Based Social Pairing Agent

---

**Require:** User query $Q$, conversation history $H$, agent personality profile $P$, maximum iterations $T$, convergence threshold $\epsilon$

1: Modules $\leftarrow$ {Emotion, Memory, Planning, SocialNorms, GoalTracking}
2: Modules $\leftarrow$ InitializeModules($P$)                    ▷ Initialize modules based on personality traits
3: GW $\leftarrow$ InitializeGlobalWorkspace($P, H$)
4: SC $\leftarrow$ InitializeSalienceCalculator($P$)
5: $R_{\text{prev}} \leftarrow \emptyset$                    ▷ Store previous module responses
6: **for** $t = 1 \rightarrow T$ **do**
7:     **parallel for** each module $M \in$ Modules:
8:         $R_M \leftarrow M.\text{PROCESS}(Q, H, \text{GW})$                    ▷ Module's local response
9:     **for all** $M \in$ Modules **do**
10:         $S_M \leftarrow \text{SC.EVALUATE}(R_M, P)$                    ▷ Compute salience score
11:     **end for**
12:     $M^* \leftarrow \arg\max_M S_M$                    ▷ Select most salient module
13:     $C \leftarrow R_{M^*}$                    ▷ Content to broadcast
14:     **if** $\exists M_i, M_j : \text{Conflict}(R_{M_i}, R_{M_j}) > $ threshold **then**
15:         $C \leftarrow \text{RESOLVECONFLICT}(R_{M_i}, R_{M_j}, \text{GW}, P)$                    ▷ Resolve module conflicts
16:     **end if**
17:     $\text{IGNITE}(\text{GW}, C)$                    ▷ Activate content in workspace
18:     $\text{BROADCAST}(C)$ to all modules
19:     **for all** $M \in$ Modules **do**
20:         $M.\text{UPDATE}(C, \text{GW})$                    ▷ Update module state
21:     **end for**
22:     **if** $\max_M |R_M - R_{\text{prev},M}| < \epsilon$ **or** $t = T$ **then**                    ▷ Minimal change in responses
23:         **break**
24:     **end if**
25:     $R_{\text{prev}} \leftarrow \{R_M | M \in \text{Modules}\}$                    ▷ Save current responses for next iteration
26: **end for**
27: Response $\leftarrow \text{INTEGRATEMODULEOUTPUTS}(\{R_M | M \in \text{Modules}\}, \text{GW}, P)$
28: Memory.UPDATELONGTERM($Q$, Response, $H$)                    ▷ Update long-term memory
29: $P \leftarrow \text{UPDATEPREFERENCES}(P, Q, \text{Response}, H)$                    ▷ Adaptively update preferences
30: **return** Response                    ▷ Return final integrated response

---

---

**Algorithm 2** CogniPair: Speed Dating Cognitive Simulation System

---

**Require:** Agent profiles $\mathcal{P} = \{P_1, P_2, \ldots, P_n\}$, batch size $b$, cognitive parameters $\Theta$
**Ensure:** Dating results $\mathcal{R}$, cognitive trace data $\mathcal{T}$, matches $\mathcal{M}$

1: $\mathcal{A} \leftarrow \emptyset$                ▷ Empty agent set
2: $\mathcal{M} \leftarrow \emptyset$                ▷ Empty matches set
3: $\mathcal{T} \leftarrow \emptyset$                ▷ Empty cognitive trace set
4: **for** $P_i \in \mathcal{P}$ **do**            ▷ Initialize all agents
5:      $\text{Modules}_i \leftarrow \{\text{Emotion, Memory, Planning, SocialNorms, Attraction}\}$
6:      $\text{GW}_i \leftarrow \text{InitializeGlobalWorkspace}(P_i)$       ▷ Initialize global workspace
7:      $\text{DA}_i \leftarrow \text{InitializeDatingAttributes}(P_i)$          ▷ Map dating preferences
8:      $A_i \leftarrow \text{CognitiveAgent}(P_i, \text{Modules}_i, \text{GW}_i, \text{DA}_i, \Theta)$
9:      $\mathcal{A} \leftarrow \mathcal{A} \cup \{A_i\}$
10: **end for**
11: $\mathcal{P}_{pairs} \leftarrow \text{GenerateCompatiblePairs}(\mathcal{A})$       ▷ Based on gender/orientation
12: $\mathcal{B} \leftarrow \text{BatchPairs}(\mathcal{P}_{pairs}, b)$            ▷ Create batches of size $b$
13: **for** batch $B \in \mathcal{B}$ **do**
14:      $\text{Results}_B \leftarrow \emptyset$
15:      **for** pair $(A_i, A_j) \in B$ **do parallel**         ▷ Process pairs in parallel
16:          $H_{ij} \leftarrow \emptyset$            ▷ Empty conversation history
17:          $C_{ij} \leftarrow \text{InitializeContext}(A_i, A_j)$       ▷ Setting, shared context
18:          **for** $r = 1 \rightarrow \text{MAX\_ROUNDS}$ **do**       ▷ Conversation rounds
19:             $Q_i \leftarrow A_i.\text{GENERATEQUERY}(H_{ij}, C_{ij})$     ▷ Generate question/statement
20:             $\mathcal{T} \leftarrow \mathcal{T} \cup A_i.\text{GW}.\text{GETTRACE}()$      ▷ Capture cognitive trace
21:             $H_{ij} \leftarrow H_{ij} \cup \{(i, Q_i)\}$      ▷ Update conversation history
22:             $R_j \leftarrow A_j.\text{GENERATERESPONSE}(Q_i, H_{ij}, C_{ij})$      ▷ Using Alg. 1
23:             $\mathcal{T} \leftarrow \mathcal{T} \cup A_j.\text{GW}.\text{GETTRACE}()$      ▷ Capture cognitive trace
24:             $H_{ij} \leftarrow H_{ij} \cup \{(j, R_j)\}$      ▷ Update conversation history
25:             **if** $r < \text{MAX\_ROUNDS}$ **then**
26:                 $Q_j \leftarrow A_j.\text{GENERATEQUERY}(H_{ij}, C_{ij})$
27:                 $\mathcal{T} \leftarrow \mathcal{T} \cup A_j.\text{GW}.\text{GETTRACE}()$
28:                 $H_{ij} \leftarrow H_{ij} \cup \{(j, Q_j)\}$
29:                 $R_i \leftarrow A_i.\text{GENERATERESPONSE}(Q_j, H_{ij}, C_{ij})$      ▷ Using Alg. 1
30:                 $\mathcal{T} \leftarrow \mathcal{T} \cup A_i.\text{GW}.\text{GETTRACE}()$
31:                 $H_{ij} \leftarrow H_{ij} \cup \{(i, R_i)\}$
32:             **end if**
33:             $A_i.\text{UPDATEATTRACTION}(H_{ij}, A_j)$      ▷ Update attraction dynamics
34:             $A_j.\text{UPDATEATTRACTION}(H_{ij}, A_i)$
35:          **end for**
36:          $S_i \leftarrow A_i.\text{EVALUATECOMPATIBILITY}(A_j, H_{ij})$      ▷ Final decision
37:          $S_j \leftarrow A_j.\text{EVALUATECOMPATIBILITY}(A_i, H_{ij})$
38:          $\text{Results}_B \leftarrow \text{Results}_B \cup \{(A_i, A_j, S_i, S_j, H_{ij})\}$
39:          **if** $S_i \geq \text{THRESHOLD} \wedge S_j \geq \text{THRESHOLD}$ **then**
40:             $\mathcal{M} \leftarrow \mathcal{M} \cup \{(A_i, A_j)\}$      ▷ Record match
41:          **end if**
42:          $A_i.\text{UPDATEPREFERENCES}(A_j, H_{ij}, S_i, S_j)$      ▷ Preference evolution
43:          $A_j.\text{UPDATEPREFERENCES}(A_i, H_{ij}, S_j, S_i)$
44:      **end for**
45:      $\mathcal{R} \leftarrow \mathcal{R} \cup \text{Results}_B$
46: **end for**
47: **return** $(\mathcal{R}, \mathcal{T}, \mathcal{M})$

---

**Algorithm 3** Adventure-Based Personality Assessment

1: Initialize `personalityProfile` and `traitConfidence`
2: `modelPreference` = GetUserModelPreference()        ▷ Cloud (GPT-4o) or Local (Ollama)
3: **while** more scenarios needed AND trait coverage insufficient **do**
4:     Select and present next scenario from pool
5:     Collect user's choice
6:     **if** `modelPreference` == "local" **then**
7:         Analyze choice using Ollama (llama3 or deepseek-r1)
8:     **else**
9:         Analyze choice using GPT-4o
10:     **end if**
11:     Update personality traits based on LLM analysis
12:     Generate follow-up question based on user's choice
13:     Collect user's free-text response
14:     **if** `modelPreference` == "local" **then**
15:         Analyze free text using Ollama
16:     **else**
17:         Analyze free text using GPT-4o
18:     **end if**
19:     Update personality traits based on text analysis
20: **end while**
21: Normalize and validate final personality profile
22: **return** `finalProfile`

## A.3 DETAILED SYSTEM FLOW ANALYSIS: GNWT-AGENT COGNITIVE ARCHITECTURE

This appendix presents a step-by-step analysis of information flow through the GNWT-Agent cognitive architecture. GNWT-Agent's central innovation is its hybrid neural-symbolic approach that combines specialized neural modules with LLM reasoning via a global workspace mechanism, implementing a neurobiologically-informed cognitive architecture.

## A.4 FORMAL ARCHITECTURE DEFINITION

The GNWT-Agent cognitive architecture is formally defined as a quintuple:

$$\mathcal{E} = (\mathcal{M}, \mathcal{W}, \mathcal{I}, \mathcal{L}, \mathcal{P}) \tag{12}$$

Where:

$\mathcal{M} = \{M_{\text{emo}}, M_{\text{mem}}, M_{\text{plan}}, M_{\text{norm}}, M_{\text{goal}}\}$ represents the set of specialized cognitive modules
$\mathcal{W}$ denotes the global workspace integration mechanism
$\mathcal{I}$ signifies the information bottleneck attention system
$\mathcal{L}$ represents the language model interface
$\mathcal{P}$ characterizes the personality representation space

Each cognitive module $M_i \in \mathcal{M}$ implements a hybrid neural-symbolic architecture:

$$M_i = (\mathcal{N}_i, \mathcal{L}_i, \mathcal{T}_i, \mathcal{D}_i, \mathcal{I}_i, \mathcal{S}_i) \tag{13}$$

Where:

$\mathcal{N}_i$ denotes the neural processing component
$\mathcal{L}_i$ signifies the module-specific LLM component
$\mathcal{T}_i$ represents the tensor-text conversion mechanism
$\mathcal{D}_i$ denotes the differentiable memory system
$\mathcal{I}_i$ signifies the module interface specification
$\mathcal{S}_i$ represents the salience computation function

Table 1: Specialized Modules and Their Neuroanatomical Bases

| Module | Function | Neural Inspiration |
| --- | --- | --- |
| Emotion ($M_{\text{emo}}$) | Affective processing | Limbic system, amygdala, insula |
| Memory ($M_{\text{mem}}$) | Information retrieval | Hippocampus, temporal cortex |
| Planning ($M_{\text{plan}}$) | Structured reasoning | Frontopolar cortex, DLPFC |
| Social Norms ($M_{\text{norm}}$) | Social context | mPFC, TPJ |
| Goal Tracking ($M_{\text{goal}}$) | Hierarchical goals | OFC, ACC |

## A.5 INFORMATION FLOW PROCESS

### A.5.1 INITIAL TEXT ENCODING

The information flow begins with the transformation of text inputs into neural representations:

$$
\begin{aligned}
e_Q &= \phi_{\text{embed}}(Q) \in \mathbb{R}^d \\
e_H &= \{\phi_{\text{embed}}(h_i) | h_i \in H\} \in \mathbb{R}^{n \times d} \\
e_P &= \psi(p) \in \mathbb{R}^{d_p}
\end{aligned}
\tag{14}
$$

Where $\phi_{\text{embed}}$ is the embedding model that converts text to dense vectors, $Q$ is the query text, $H$ is the conversation history, and $p$ is the personality profile.

### A.5.2 FEATURE EXTRACTION AND EMBEDDING ALIGNMENT

The raw embeddings undergo feature extraction and alignment:

$$
\begin{aligned}
e'_Q &= \text{FeatureExtractor}(e_Q) \\
e'_H &= \{\text{FeatureExtractor}(e_h) | e_h \in e_H\}
\end{aligned}
\tag{15}
$$

### A.5.3 MODULE SALIENCE CALCULATION

Each module calculates its relevance to the current input:

$$
\begin{aligned}
s_i &= \mathcal{S}_i(e'_Q, e'_H, e_P, \mathcal{G}) \\
&= \alpha_i + \sum_j \beta_{ij} \cdot f_{ij}(e'_Q, e'_H, e_P, \mathcal{G})
\end{aligned}
\tag{16}
$$

Where $\alpha_i$ is the baseline salience, $\beta_{ij}$ are weighting coefficients, $f_{ij}$ are feature extractors, and $\mathcal{G}$ is the current global workspace state.

### A.5.4 PARALLEL MODULE PROCESSING

Each module independently processes the input using a hybrid neural-LLM approach with the following general pattern:

1. **Neural Processing**: Extract relevant features and apply module-specific transformations
2. **Tensor-to-Text Conversion**: Convert neural representations to LLM-readable format
3. **LLM Processing**: Generate structured symbolic representations using prompting
4. **Text-to-Tensor Conversion**: Transform LLM outputs back to neural representations
5. **Output Integration**: Combine neural and symbolic components for module output

Module-specific processing includes:

- **Emotion Module**: Detects emotional states and regulation strategies
- **Memory Module**: Retrieves and integrates episodic, semantic, and working memories
- **Planning Module**: Generates structured plans for achieving identified goals
- **Social Norms Module**: Identifies appropriate social contexts and behavioral norms
- **Goal Tracking Module**: Maintains hierarchical goal representations and tracks progress

### A.5.5 GLOBAL WORKSPACE INTEGRATION

Module outputs are projected to a common workspace dimension and integrated based on salience:

### A.5.6 FINAL PROMPT CONSTRUCTION

The global workspace state informs the construction of the final prompt for the response LLM, transforming module outputs into a coherent instruction format:

$$
\begin{aligned}
P_{\text{system}} = \text{JoinWithNewlines}(&\text{PersonalityPrompt}(e_P), \\
&\text{ModulePrompts}(\{\text{output}_i\}, \{\text{adjusted\_weights}_i\}), \\
&\text{StrategyPrompt}(\text{integrated\_state}), \\
&\text{ConflictPrompt}(\text{conflicts\_resolved}), \\
&\text{ResponsePlanPrompt}(\text{response\_plan}))
\end{aligned}
\tag{17}
$$

---

**Algorithm 4** Global Workspace Integration

---

1: $\text{module\_rep}_i \leftarrow \text{ProjectToWorkspace}(\text{output}_i)$ for each module $i$
2: $\text{normalized\_salience} \leftarrow \text{Softmax}(s)$
3: $\text{personality\_weights} \leftarrow \text{Softmax}(\text{PersonalityToIntegration}(e_P))$
4: $\text{combined\_weights} \leftarrow \frac{\text{normalized\_salience} + \text{personality\_weights}}{2}$
5: **for** each pair of modules $(i, j)$ where $i \neq j$ **do**
6:     $\text{conflict\_scores}_{i,j} \leftarrow \text{ConflictDetector}([\text{module\_rep}_i, \text{module\_rep}_j])$
7: **end for**
8: Apply conflict resolution adjustments to weights
9: $\text{integrated\_output} \leftarrow \sum_i \text{adjusted\_weights}_i \cdot \text{module\_rep}_i$
10: **if** major conflicts detected **then**
11:     Apply specialized conflict resolution
12: **end if**
13: $\text{workspace\_state} \leftarrow \text{WorkspaceProjection}(\text{integrated\_output})$
14: Perform integration LLM call to further refine integration

---

### A.5.7 RESPONSE GENERATION AND MEMORY UPDATE

The final response is generated and memory systems are updated:

$$
\begin{aligned}
\text{history} &= [\text{prev\_messages}] \\
\text{query\_message} &= \{\text{"role": "user", "content": } Q\} \\
\text{full\_history} &= \text{history } + [\text{query\_message}] \\
R &= \text{LLM}_{\text{response}}(P_{\text{system}}, \text{full\_history}, \text{temperature} = 0.7)
\end{aligned}
\tag{18}
$$

$$
\begin{aligned}
&\text{EpisodicMemory.write}(\text{Embed}(Q, R), \text{StructuredEncoding}(Q, R, \text{workspace\_state})) \\
&\text{WorkingMemory.update}(Q, R)
\end{aligned}
\tag{19}
$$

### A.6 PROCESSING EXAMPLE

The following example demonstrates the architecture's response to: "I'm really nervous about my job interview tomorrow. I've been preparing for weeks, but I still don't feel ready. What should I do?"

---

**Module Salience Calculation**

**Salience Scores:**

- Emotion Module: 0.75 (anxiety markers, moderate neuroticism, emotional shift)
- Planning Module: 0.67 (clear goal, structured task, high conscientiousness)
- Memory Module: 0.52 (relevant experiences, moderate openness)
- Social Norms Module: 0.48 (professional context, formal situation)
- Goal Tracking Module: 0.45 (career advancement, interview milestone)

**Normalized weights:** [0.31, 0.28, 0.21, 0.12, 0.08]

---

**Global Workspace Integration**

**Conflict Detection:**

- Primary conflict: Emotion module (anxiety validation) vs. Planning module (action steps)

**Adjusted Weights:** [0.29, 0.25, 0.22, 0.14, 0.10]

**Integration Results:**

- Primary response focus: balanced emotional and practical support
- Emotional context: anxiety (valence: -0.48, arousal: 0.68) with determination
- Response strategy: validate anxiety then provide structured help
- Conflict resolution: Balance validation with actionable steps

**Response Plan:**

1. Validate interview anxiety as normal
2. Acknowledge preparation effort positively
3. Provide targeted high-impact preparation steps
4. Include confidence-building perspective
5. End with encouragement and well-wishes

**Final Response**

It's completely normal to feel nervous before an interview, especially for a company you're really excited about! The fact that you've been preparing for weeks shows your dedication, and that preparation won't disappear overnight, even if you're feeling anxious right now.

At this point, here's what I'd suggest focusing on:

1. Review your strongest points - skim through the key qualifications and 2-3 accomplishment stories you want to highlight. Don't try to memorize everything.

2. Practice answering 3-5 common questions out loud - this builds muscle memory that helps even when nerves kick in.

3. Prepare your logistics for tomorrow - lay out your outfit, plan your route, and gather any materials you need to bring.

4. Most importantly, make sure you get adequate rest tonight. A clear mind will serve you better than a few more hours of anxious preparation.

Remember that interviews are a two-way conversation, not an examination. You've been preparing for weeks - now it's time to trust that preparation and focus on presenting your authentic self.

Good luck tomorrow! You've put in the work, and that's the most important part.

## A.7 MODULE LLM CALL ARCHITECTURE

The GNWT-Agent architecture employs a cascade of LLM calls with neural pre-processing and post-processing:

$$\text{Module-Level LLM Calls} : \{\text{LLM}_{\text{emo}}(P_{\text{emo}}), \text{LLM}_{\text{mem}}(P_{\text{mem}}), \text{LLM}_{\text{plan}}(P_{\text{plan}}),$$
$$\text{LLM}_{\text{norm}}(P_{\text{norm}}), \text{LLM}_{\text{goal}}(P_{\text{goal}})\} \tag{20}$$

$$\text{Integration LLM Call} : \text{LLM}_{\text{integration}}(P_{\text{integration}}) \tag{21}$$

$$\text{Response LLM Call} : \text{LLM}_{\text{response}}(P_{\text{system}}, Q) \tag{22}$$

This creates a complete cognitive cycle:

$$\text{Input} \rightarrow \text{Neural Processing} \rightarrow \text{Module LLMs} \rightarrow \text{Integration LLM} \rightarrow \text{Response LLM} \rightarrow \text{Memory Update} \tag{23}$$

### A.8 Summary

The GNWT-Agent cognitive architecture implements a hybrid neural-symbolic approach that integrates multiple specialized modules within a neurobiologically-informed framework. The architecture's core components include five specialized cognitive modules addressing emotion, memory, planning, social norms, and goals.

The information flow through the system follows a comprehensive process:

1. Initial text encoding via dense vector representations
2. Parallel processing across specialized modules
3. Dynamic salience-based resource allocation
4. Global workspace integration with conflict detection and resolution
5. Structured prompt construction
6. Final response generation
7. Memory update and continuous learning

Table 2: Strengths and Limitations of GNWT-Agent Architecture

| Strengths | Limitations |
|---|---|
| Enhanced interpretability through explicit module contributions | Computational complexity due to multiple LLM calls |
| Improved uncertainty handling via Bayesian uncertainty propagation | Challenges in consistent knowledge representation |
| Consistent personality representation through dedicated space | Need for further empirical validation |
| Dynamic adaptation through salience-based processing | Potential for response latency issues |
| Neurobiologically-inspired cognitive processing | Complex architecture requiring extensive fine-tuning |

In conclusion, the GNWT-Agent cognitive architecture represents a significant step toward creating AI systems that not only process language effectively but do so through mechanisms that more closely approximate human cognitive processes, potentially leading to more natural, adaptive, and comprehensible AI interactions.

## A.9 WORKED EXAMPLE: INTERVIEW ANXIETY QUERY

This section presents a detailed step-by-step walkthrough of the complete GNWT-Agent cognitive architecture processing flow for a single example query. By tracing the transformations from raw input to final response, we provide a concrete illustration of the theoretical architecture described in previous sections.

## A.10 EXAMPLE INPUT QUERY

The example query represents a common scenario of pre-interview anxiety:

> **User Input**
>
> "I'm really nervous about my job interview tomorrow. I've been preparing for weeks, but I still don't feel ready. What should I do?"

## A.11 INITIAL TEXT ENCODING AND EMBEDDING

The first transformation converts the raw text into numerical representations:

$$\text{Input} \xrightarrow{\phi_{\text{embed}}} \text{Query Embedding} \tag{24}$$

$$\text{"I'm really nervous..."} \to [0.086, -0.143, 0.257, \dots, 0.112] \in \mathbb{R}^{768} \tag{25}$$

Previous conversation context is also encoded:

$$\text{"I've been job hunting for months"} \to [0.141, 0.092, \dots] \in \mathbb{R}^{768} \tag{26}$$

$$\text{"I finally got an interview at my dream company"} \to [0.235, -0.124, \dots] \in \mathbb{R}^{768} \tag{27}$$

Feature extraction enhances these raw embeddings:

$$\text{Enhanced query embedding: } [0.127, -0.086, 0.313, \dots] \in \mathbb{R}^{768} \tag{28}$$

## A.12 MODULE SALIENCE CALCULATION

Each module calculates its relevance to the query through multi-factor salience functions:

| Module | Primary Factors | Values | Raw Salience | Normalized |
|--------|-----------------|--------|--------------|------------|
| Emotion | Emotional intensity | 0.83 | 0.75 | 0.31 |
| | Personality factor | 0.52 | | |
| | Emotional change | 0.60 | | |
| Planning | Goal clarity | 0.78 | 0.67 | 0.28 |
| | Task structure | 0.82 | | |
| | Personality factor | 0.76 | | |
| Memory | Memory match | 0.64 | 0.52 | 0.21 |
| | Personality factor | 0.61 | | |
| | Recency boost | 0.35 | | |
| Social Norms | Norm relevance | 0.71 | 0.48 | 0.12 |
| | Formality | 0.70 | | |
| | Personality factor | 0.63 | | |
| Goal Tracking | Goal relevance | 0.76 | 0.45 | 0.08 |
| | Personality factor | 0.65 | | |
| | Progress factor | 0.58 | | |

Table 3: Module salience calculation for the interview anxiety query

The emotional content of the query results in the Emotion Module having the highest salience (0.31), followed closely by the Planning Module (0.28).

## A.13 PARALLEL MODULE PROCESSING

### A.13.1 EMOTION MODULE PROCESSING

The Emotion Module performs neural processing followed by LLM reasoning:

Figure 7: Emotion Module processing flow

### A.13.2 PLANNING MODULE PROCESSING

The Planning Module extracts the goal and generates structured steps:

```
Planning Module LLM Output

{
  "requires_planning": true,
  "planning_type": "preparation_strategy",
  "steps": [
    {
      "content": "Review core interview preparation materials",
      "confidence": 0.93
    },
    {
      "content": "Practice answering common questions aloud",
      "confidence": 0.89
    },
    {
      "content": "Prepare concise examples of achievements",
      "confidence": 0.86
    },
    {
      "content": "Implement anxiety reduction techniques",
      "confidence": 0.82
    },
    {
      "content":
      "Plan interview logistics (route, outfit, materials)",
      "confidence": 0.91
    },
    {
      "content": "Get adequate rest before interview",
      "confidence": 0.85
    }
  ],
  "plan_feasibility": 0.88,
  "goal_alignment": 0.84,
  "plan_uncertainty": 0.16
}
```

### A.13.3 MEMORY MODULE PROCESSING

The Memory Module retrieves relevant episodic and semantic memories:

```
Memory Module Content

{
  "episodic_memories": [
    {"content": "User mentioned job hunting for months",
    "confidence": 0.88},
    {"content": "User expressed excitement about interview at
    dream company",
     "confidence": 0.92},
    {"content":
    "User indicated spending significant time preparing",
     "confidence": 0.85}
  ],
  "semantic_knowledge": [
    {"content": "Interview anxiety is normal and expected",
    "confidence": 0.94},
    {"content": "Over-preparation can sometimes increase anxiety",
     "confidence": 0.82},
    {"content": "Sleep quality affects interview performance",
    "confidence": 0.89}
  ]
}
```

### A.13.4   SOCIAL NORMS MODULE PROCESSING

The Social Norms Module identifies the appropriate context and behavioral expectations:

```
Social Norms Module Content

{
  "social_context": "professional_development",
  "appropriate_norms": [
    "acknowledge interview anxiety as normal",
    "validate preparation efforts",
    "provide practical confidence-building advice",
    "offer both emotional and practical support",
    "maintain appropriate emotional engagement"
  ],
  "formality_level": 0.65,
  "appropriate_tone": "supportive_professional"
}
```

### A.13.5   GOAL TRACKING MODULE PROCESSING

The Goal Tracking Module identifies and relates current goals to existing goals:

```
Goal Tracking Module Content

{
  "identified_goals": [
    {"content": "Successfully complete job interview",
    "explicitness": 0.92},
    {"content":
    "Reduce pre-interview anxiety", "explicitness": 0.85}
  ],
  "existing_goals_relevance": [
    {"goal": "Secure desired employment", "relevance": 0.94},
    {"goal":
    "Develop career in chosen field", "relevance": 0.86}
  ]
}
```

## A.14 GLOBAL WORKSPACE INTEGRATION

The outputs from all five modules are integrated through the global workspace mechanism:

### A.14.1 CONFLICT DETECTION AND RESOLUTION

The system identifies and resolves potential conflicts between module outputs:

|  | Emotion | Planning | Memory | Social | Goal |
|---|---|---|---|---|---|
| Emotion | 0.00 | **0.35** | 0.10 | 0.05 | 0.10 |
| Planning | **0.35** | 0.00 | 0.15 | 0.10 | 0.05 |
| Memory | 0.10 | 0.15 | 0.00 | 0.05 | 0.10 |
| Social | 0.05 | 0.10 | 0.05 | 0.00 | 0.05 |
| Goal | 0.10 | 0.05 | 0.10 | 0.05 | 0.00 |

Table 4: Conflict detection matrix showing highest conflict between Emotion and Planning modules

The primary conflict (0.35) exists between the Emotion Module's emphasis on validation and the Planning Module's focus on action steps. This conflict affects the modules' weights:

$$\text{Original weights: } [0.31, 0.28, 0.21, 0.12, 0.08] \tag{29}$$
$$\text{Conflict-adjusted weights: } [0.29, 0.25, 0.22, 0.14, 0.10] \tag{30}$$

### A.14.2 INTEGRATION RESULT

The Integration LLM synthesizes information from all modules into a coherent state:

```
Integration LLM Output (Abbreviated)

{
  "integrated_state": {
    "primary_response_focus": "balanced_emotional_practical",
    "emotional_context": {
      "emotion": "anxiety",
      "secondary": "determination",
      "valence": -0.48,
      "arousal": 0.68
    },
    "response_strategy": {
      "approach":
      "validate_anxiety_then_provide_structured_help",
      "tone": "supportive_professional",
      "formality": "moderate"
    },
    "response_plan": [
      "validate interview anxiety as normal",
      "acknowledge preparation effort positively",
      "provide targeted high-impact preparation steps",
      "include confidence-building perspective",
      "end with encouragement and well-wishes"
    ]
  },
  "conflicts_resolved": [
    {
      "conflict": "Emotion module emphasizes validation vs
      Planning module focuses on action steps",
      "resolution": "Balance emotional validation with
      practical action steps by
                    first acknowledging feelings then providing
                    concrete,
                    manageable next steps",
      "confidence": 0.88
    }
  ]
}
```

A.15  FINAL PROMPT CONSTRUCTION

The integrated state is transformed into a structured system prompt for the response generation LLM:

**System Prompt**

```
You are a cognitive agent with the following personality:
MBTI Type: ENFJ
Big 5 Traits:
- Openness: high (0.70)
- Conscientiousness: high (0.80)
- Extraversion: moderate (0.50)
- Agreeableness: very high (0.90)
- Neuroticism: moderate-low (0.40)

[EMOTION MODULE] (Contribution: 0.29): Pay careful attention to
the anxiety and nervousness the user is experiencing about
their upcoming job interview. The emotions detected include
high anxiety (0.82) with a negative valence (-0.48) but also
determination (0.55). Emotional content identified: "really
nervous", "don't feel ready". Use a validation_with_reframing
approach that acknowledges the anxiety while gently building
confidence.

[PLANNING MODULE] (Contribution: 0.25):
Provide structured, preparation_strategy advice for the user's
interview preparation. Focus on high-impact preparation
activities that can be completed in limited time. Consider
including: reviewing core materials, practicing answers aloud,
preparing achievement examples, reducing anxiety, handling
logistics, and getting rest.

[MEMORY MODULE] (Contribution: 0.22):
Reference these key points from conversation history:
User has been job hunting for months,  this is an interview at
their "dream company," and they have already spent significant
time preparing. Also incorporate knowledge that interview
anxiety is normaland that over-preparation can sometimes
increase anxiety rather than helping.

[SOCIAL NORMS MODULE] (Contribution: 0.14):
Maintain a supportive_professional
tone appropriate for career_development context.
Acknowledge that interview
anxiety is normal while validating preparation efforts.
Observe these norms:
validate emotions, show empathy, offer practical support,
maintain appropriate emotional engagement.

Follow this response strategy:
validate_anxiety_then_provide_structured_help with
a supportive_professional tone at moderate formality level.

When addressing conflicts:
Balance emotional validation with practical action steps
by first acknowledging feelings then providing concrete,
manageable next steps.

Process the query by considering each
cognitive dimension above according to
its contribution weight. Follow this response plan:
1. Validate interview anxiety as normal
2. Acknowledge preparation effort positively
3. Provide targeted high-impact preparation steps
4. Include confidence-building perspective
5. End with encouragement and well-wishes
```

## A.16 RESPONSE GENERATION

The Response LLM generates the final output based on the system prompt:

> **Generated Response**
>
> It's completely normal to feel nervous before an interview, especially for a company you're really excited about! The fact that you've been preparing for weeks shows your dedication, and that preparation won't disappear overnight, even if you're feeling anxious right now.
> At this point, here's what I'd suggest focusing on:
> 1. Review your strongest points - skim through the key qualifications and 2-3 accomplishment stories you want to highlight. Don't try to memorize everything.
> 2. Practice answering 3-5 common questions out loud - this builds muscle memory that helps even when nerves kick in.
> 3. Prepare your logistics for tomorrow - lay out your outfit, plan your route, and gather any materials you need to bring.
> 4. Most importantly, make sure you get adequate rest tonight. A clear mind will serve you better than a few more hours of anxious preparation.
> Remember that interviews are a two-way conversation, not an examination. You've been preparing for weeks - now it's time to trust that preparation and focus on presenting your authentic self.
> Good luck tomorrow! You've put in the work, and that's the most important part.

## A.17 INFORMATION FLOW ANALYSIS

Examining the example reveals several key aspects of the GNWT-Agent architecture:

1. **Modular Specialization**: Each module contributes distinct information:
   - Emotion Module provides emotional understanding and regulation strategy
   - Planning Module offers concrete action steps
   - Memory Module contributes contextual information and relevant knowledge
   - Social Norms Module guides appropriate tone and interaction norms
   - Goal Tracking Module ensures alignment with long-term objectives

2. **Salience-Based Processing**: The architecture dynamically allocates attention based on query characteristics, with emotional content receiving the highest weight in this anxiety-focused query.

3. **Conflict Resolution**: The system explicitly identifies and resolves the tension between emotional validation and practical advice through a balanced approach.

4. **Structured Response Planning**: The final response follows the five-step plan specified in the integration phase, creating a cohesive structure that addresses multiple dimensions of the query.

5. **Personality Influence**: The system's responses reflect the specified personality characteristics, particularly high agreeableness (0.90) through the empathetic tone.

The response demonstrates how the GNWT-Agent architecture produces outputs that balance emotional responsiveness with practical utility, organized through a structured cognitive framework that mimics aspects of human cognition.

### A.18 Use Case: Dating Application

### A.19 Overview

CogniPair for Dating represents a novel approach to matchmaking that leverages the GNWT-Agent cognitive architecture to create realistic digital twins of users. Unlike traditional dating platforms that rely on static profiles and rule-based matching algorithms, CogniPair simulates genuine cognitive interactions between potential matches before they ever meet in person. This system models personality traits, emotional responses, memory formation, social norm adherence, and planning capabilities to predict compatibility with high accuracy.

### A.20 User Journey

The CogniPair Dating platform guides users through the following process:

1. **Profile Creation:** Users complete an in-depth personality assessment and preference questionnaire that captures traits across multiple dimensions including the Big Five personality factors.

2. **Digital Twin Generation:** The GNWT-Agent architecture constructs a cognitive digital twin that faithfully reproduces the user's conversational patterns, emotional responses, and decision-making processes.

3. **Simulated Interactions:** Digital twins engage in mock conversations with potential matches, processing information through specialized cognitive modules.

4. **Compatibility Analysis:** The system analyzes cognitive states throughout conversations, identifying mutual interests, conversational flow, emotional resonance, and decision-making alignment.

5. **Match Recommendation:** Users receive matches based on authentic cognitive compatibility rather than superficial profile similarities.

### A.21 Case Study: Emma and Michael

To illustrate CogniPair's capabilities, we present a case study of two users, Emma and Michael, whose digital twins engaged in a simulated conversation.

#### A.21.1 User Profiles

| Attribute | Emma | Michael |
|---|---|---|
| **Age** | 28 | 31 |
| **Background** | Marketing director, enjoys outdoor activities, travel | Software engineer, passionate about music, cooking, technology |
| **Personality Traits** | Openness: 0.75
Conscientiousness: 0.62
Extraversion: 0.70
Agreeableness: 0.82
Neuroticism: 0.35 | Openness: 0.68
Conscientiousness: 0.82
Extraversion: 0.45
Agreeableness: 0.74
Neuroticism: 0.42 |
| **Dating Attributes** | Attractiveness: 8.2/10
Sincerity: 8.7/10
Intelligence: 8.5/10
Fun: 8.9/10
Ambition: 8.8/10
Shared Interests: 7.9/10 | Attractiveness: 7.8/10
Sincerity: 9.1/10
Intelligence: 9.0/10
Fun: 7.5/10
Ambition: 8.7/10
Shared Interests: 8.2/10 |

Table 5: Profiles of Emma and Michael with personality traits and dating attributes

A.21.2    SIMULATED CONVERSATION WITH COGNITIVE PROCESSING

The following transcript shows the simulated conversation between Emma and Michael's digital twins, with real-time cognitive processing visualized.

**System Introduction:** Welcome to a speed date! This is a first conversation between two people who are considering dating. Please introduce yourselves.

**Emma:** Hi there! I'm Emma. I work in marketing and I'm really passionate about creating campaigns that connect with people. In my free time, I love hiking and being outdoors whenever possible. I also try to travel to at least one new country each year. What about you?

---

**Emma's Cognitive Processing**

**Module Activations:**

- **Emotion Module: 0.57** - Moderate activation showing enthusiasm and openness
- **Social Norms Module: 0.64** - High activation ensuring appropriate self-disclosure
- **Memory Module: 0.35** - Low activation retrieving personal information
- **Planning Module: 0.31** - Low activation structuring introduction
- **Goal Tracking Module: 0.42** - Moderate activation focusing on relationship exploration

---

**Michael:** Nice to meet you, Emma! I'm Michael. I work as a software engineer, focusing on user experience design. I find it fascinating to solve problems that make technology more intuitive. Outside of work, I'm really into music—I play guitar and go to concerts whenever I can. I also love experimenting with cooking, especially dishes from places I've never been to. Your travel experiences sound amazing. What's been your favorite country to visit so far?

---

**Michael's Cognitive Processing**

**Module Activations:**

- **Emotion Module: 0.45** - Moderate activation showing interest
- **Social Norms Module: 0.55** - Moderate activation ensuring appropriate response
- **Memory Module: 0.38** - Low-moderate activation retrieving personal details
- **Planning Module: 0.51** - Moderate activation structuring response and question
- **Goal Tracking Module: 0.49** - Moderate activation exploring compatibility

---

**Emma:** It's nice to meet someone who appreciates music! I'm not musically talented myself, but I love going to live shows. As for travel, that's a tough question! I'd say Japan was probably the most fascinating—the blend of traditional culture with cutting-edge technology was incredible. Plus, the food was amazing. Do you enjoy traveling yourself, or is cooking your way of experiencing different cultures?

> **Emma's Cognitive Processing**
>
> **Module Activations:**
>
> - **Emotion Module: 0.68** - High activation showing increased enthusiasm
> - **Memory Module: 0.72** - High activation recalling travel experiences
> - **Social Norms Module: 0.53** - Moderate activation maintaining conversation flow
> - **Planning Module: 0.29** - Low activation
> - **Goal Tracking Module: 0.59** - Moderate-high activation exploring common interests
>
> **Cognitive State Analysis:** Emma's emotion module showed significant activation when discussing travel and responding to Michael's interests in music, indicating genuine enthusiasm. The memory module activated strongly when recalling specific details about Japan, suggesting authentic experiences rather than generic responses.

**Michael:** That's exactly what fascinates me about Japan too! The contrast between ancient temples and robot restaurants in Tokyo is mind-blowing. I haven't been there yet, but it's at the top of my list. I do enjoy traveling when I can, though not as regularly as you. Cooking is definitely my way of exploring cultures when I can't travel there—I recently got into making ramen from scratch after watching a documentary about Japanese food. What kind of music do you enjoy at live shows? Any favorite concerts?

> **Michael's Cognitive Processing**
>
> **Module Activations:**
>
> - **Emotion Module: 0.79** - High activation showing excitement about shared interest
> - **Memory Module: 0.68** - High activation connecting to personal experiences
> - **Social Norms Module: 0.51** - Moderate activation
> - **Planning Module: 0.35** - Low-moderate activation
> - **Goal Tracking Module: 0.73** - High activation pursuing connection point
>
> **Cognitive State Analysis:** Michael's emotion module showed strong activation when responding to the shared interest in Japan, indicating genuine excitement. The goal tracking module also activated significantly, suggesting he sees this common interest as a potential relationship foundation. His follow-up question about music demonstrates the goal-directed behavior of exploring additional shared interests.

[Conversation continues...]

### A.21.3 COMPATIBILITY DECISION PHASE

After the simulated conversation, each digital twin makes an independent compatibility assessment based on their cognitive processing throughout the interaction.

**System:** Based on your conversation, would you like to meet this person for a real date?

**Emma's Decision:** Yes, I would like to meet Michael. Our conversation felt natural and engaging. We share interests in travel and cultural experiences, and I appreciate his passion for music and cooking. His thoughtful responses and questions showed genuine interest. There's enough similarity for connection but enough difference to keep things interesting.

> **Emma's Decision Cognitive Processing**
>
> **Module Contributions to Decision:**
>
> - **Emotion Module: 0.75** - Strong positive emotional response throughout conversation
> - **Memory Module: 0.68** - High recall of conversation details and identified commonalities
> - **Social Norms Module: 0.42** - Moderate assessment of social compatibility
> - **Planning Module: 0.51** - Moderate assessment of future potential
> - **Goal Tracking Module: 0.71** - High alignment with relationship goals

**Michael's Decision:** Yes, I'm interested in meeting Emma for a real date. We have complementary interests and her enthusiasm for travel and cultural experiences resonates with me. I enjoyed how she engaged with my interests in music and cooking while sharing her own perspectives. Our conversation had a good balance of questions and sharing, and I'd like to learn more about her.

> **Michael's Decision Cognitive Processing**
>
> **Module Contributions to Decision:**
>
> - **Emotion Module: 0.69** - Strong positive emotional response
> - **Memory Module: 0.64** - Solid recall of shared interests and conversation points
> - **Social Norms Module: 0.48** - Moderate assessment of social compatibility
> - **Planning Module: 0.66** - High consideration of relationship potential
> - **Goal Tracking Module: 0.72** - High alignment with relationship goals

**Match Result:** COMPATIBLE

### A.22 COGNITIVE MODULE ACTIVITY ANALYSIS

The module activation patterns throughout the dating conversation revealed several key insights:

| Module | Introduction | Response | Follow-up | Shared Interest | Decision |
|---|---|---|---|---|---|
| Emotion | 0.57 | 0.45 | 0.68 | 0.79 | 0.72 |
| Memory | 0.35 | 0.38 | 0.72 | 0.68 | 0.66 |
| Social Norms | 0.64 | 0.55 | 0.53 | 0.51 | 0.45 |
| Planning | 0.31 | 0.51 | 0.29 | 0.35 | 0.58 |
| Goal Tracking | 0.42 | 0.49 | 0.59 | 0.73 | 0.71 |

Table 6: Cognitive module activation patterns throughout the dating conversation

The cognitive architecture revealed several key insights during this match:

- **Emotion Module:** Showed increasing activation as shared interests were discovered, peaking during discussion of travel experiences.

- **Memory Module:** Activation increased significantly when recalling specific experiences, demonstrating authentic engagement rather than superficial responses.

- **Social Norms Module:** Started high during initial introductions and gradually decreased as conversation became more comfortable.

- **Planning Module:** Peaked during the decision phase when considering future interaction potential.

- **Goal Tracking Module:** Showed steady increase throughout the conversation as relationship compatibility was assessed.

## A.23 OUTCOMES AND USER INTERFACE

After the cognitive simulation concludes, CogniPair presents users with compatible matches along with insights derived from the digital twin interactions. The platform highlights specific points of connection and potential conversation starters based on shared interests.

**CogniPair Match Report**

**Match Result:** Emma and Michael - 87% Compatibility

**Compatibility Breakdown:**

- Emotional Connection: 83%
- Conversation Flow: 91%
- Shared Interests: 78%
- Value Alignment: 85%
- Long-term Potential: 84%

**Connection Points:**

- Travel experiences (particularly interest in Japan)
- Appreciation for cultural exploration
- Complementary interests (Emma's outdoor activities, Michael's cooking)
- Similar communication styles with thoughtful questions

**Suggested Conversation Starters:**

- "I'd love to hear more about that documentary on Japanese food that inspired your cooking."
- "What's been your favorite live music experience? I'm always looking for new artists."
- "Would you want to do cooking and hiking as combined activities? Maybe prepare a meal after a trail?"

## A.24 TECHNICAL IMPLEMENTATION HIGHLIGHTS

The dating scenario leverages several key aspects of the GNWT-Agent architecture:

1. **Emotion Module Prominence:** Dating interactions show heightened emotion module activation compared to other scenarios, particularly in response to shared interests and values.
2. **Memory-Emotion Integration:** The architecture demonstrates how memories trigger emotional responses in social contexts, creating authentic patterns of engagement.
3. **Goal-Directed Decision Making:** As the conversation progresses, goal tracking module activation increases, culminating in the compatibility decision.
4. **Personal Value Assessment:** The architecture evaluates alignment in preferences, interests, and communication styles to determine overall compatibility.

This implementation demonstrates how cognitive modeling can transcend the limitations of traditional profile-based matching by simulating the nuanced psychological aspects of human connection.

## A.25 USE CASE: JOB APPLICATION

## A.26 OVERVIEW

CogniPair for Job Hiring represents a transformative approach to talent acquisition that applies the GNWT-Agent cognitive architecture to create digital twins of both hiring managers and job candidates. This system moves beyond traditional resume screening and interview processes by modeling the complex cognitive dynamics that determine professional compatibility. By simulating realistic interview interactions, the platform assesses technical skills, problem-solving approaches, communication styles, and cultural alignment before any in-person interviews take place.

## A.27 USER JOURNEY

The CogniPair Hiring platform guides users through the following process:

1. **Employer Profile Creation:** Companies input detailed job descriptions, required competencies, team dynamics, and organizational culture attributes.

2. **Candidate Profile Submission:** Candidates upload resumes and complete competency assessments that capture both technical skills and cognitive-behavioral traits.

3. **Digital Twin Generation:** The GNWT-Agent architecture creates cognitive models of both hiring managers and candidates, incorporating decision-making patterns, communication styles, and problem-solving approaches.

4. **Simulated Interviews:** Digital twins engage in multi-stage interview simulations assessing technical skills, problem-solving abilities, and cultural alignment.

5. **Compatibility Analysis:** The system analyzes cognitive states throughout the simulated interviews, identifying communication effectiveness, problem-solving compatibility, and potential team integration.

6. **Match Recommendation:** Both employers and candidates receive compatibility assessments that predict professional success and team dynamics.

## A.28 CASE STUDY: TECHSOLUTIONS INC. AND CANDIDATE ALEX

To illustrate CogniPair's capabilities in hiring contexts, we present a case study of a software development position at TechSolutions Inc. and a potential candidate, Alex.

### A.28.1 JOB AND CANDIDATE PROFILES

### A.28.2 SIMULATED INTERVIEW WITH COGNITIVE PROCESSING

The following transcript shows the simulated interview between Sarah (Engineering Manager) and Alex (Candidate), with real-time cognitive processing visualized.

**System Introduction:** This is a technical interview for a Senior Software Developer position at TechSolutions Inc.

**Sarah:** Hello Alex, thanks for joining us today. I'm Sarah, the Engineering Manager at TechSolutions. Before we dive into technical questions, could you tell me about your experience with React and how you've implemented it in previous projects?

| Attribute | TechSolutions Inc. | Alex (Candidate) |
|---|---|---|
| Position | Senior Software Developer
Team of 8, collaborative environment | 5 years software development
experience, previous team lead |
| Requirements/
Skills | JavaScript/React expertise
CI/CD pipeline experience
Agile development methodology
Problem-solving abilities | Strong JavaScript/React skills
CI/CD experience with GitLab
Certified Scrum Master
Bachelor's in Computer Science |
| Interviewer
Personality
Traits | Sarah (Engineering Manager)
Openness: 0.72
Conscientiousness: 0.86
Extraversion: 0.58
Agreeableness: 0.65
Neuroticism: 0.32 | Openness: 0.85
Conscientiousness: 0.69
Extraversion: 0.47
Agreeableness: 0.78
Neuroticism: 0.39 |
| Professional
Attributes | Technical skills: 9.2/10
Communication: 8.5/10
Leadership potential: 9.0/10
Problem-solving: 9.3/10
Cultural alignment: 8.7/10 | Technical skills: 8.8/10
Communication: 8.7/10
Leadership potential: 8.4/10
Problem-solving: 9.1/10
Cultural alignment: ?/10 (to be assessed) |
| Company
Culture/
Values | Innovative, fast-paced
Work-life balance emphasis
Continuous learning encouraged | Previous experience in fintech
Values collaborative environments
Mentor to junior developers |

Table 7: Job description and candidate profile with professional attributes

---

**Sarah's Cognitive Processing**

**Module Activations:**

- **Planning Module: 0.78** - High activation structuring interview strategy
- **Social Norms Module: 0.63** - Moderate-high activation ensuring professional tone
- **Memory Module: 0.55** - Moderate activation recalling job requirements
- **Goal Tracking Module: 0.69** - High activation focusing on technical assessment
- **Emotion Module: 0.32** - Low activation maintaining neutral assessment stance

---

**Alex:** Thank you for having me, Sarah. I've been working with React for about four years now. In my current role at FinTech Solutions, I led the migration of our legacy dashboard to a React-based architecture. One of the challenges we faced was maintaining state across a complex application with multiple user roles. I implemented a Redux state management solution with middleware for API interactions that reduced loading times by 40% and significantly improved the developer experience for my team. I also created a reusable component library that we eventually open-sourced within the company.

---

**Alex's Cognitive Processing**

**Module Activations:**

- **Memory Module: 0.82** - Very high activation retrieving specific professional experiences
- **Planning Module: 0.74** - High activation structuring response with relevant details
- **Social Norms Module: 0.61** - Moderate-high activation ensuring professional presentation
- **Goal Tracking Module: 0.69** - High activation focusing on demonstrating expertise
- **Emotion Module: 0.38** - Low-moderate activation showing appropriate enthusiasm

**Sarah:** That's impressive work with the state management solution. I'm curious about how you approached testing for that React application, especially with the Redux integration. Could you walk me through your testing strategy and any particular challenges you encountered?

---

**Sarah's Cognitive Processing**

**Module Activations:**
- **Planning Module: 0.83** - Very high activation probing technical depth
- **Memory Module: 0.76** - High activation processing previous answer against requirements
- **Goal Tracking Module: 0.72** - High activation assessing technical competency
- **Social Norms Module: 0.48** - Moderate activation
- **Emotion Module: 0.36** - Low-moderate activation showing positive response to quality answer

**Cognitive State Analysis:** Sarah's planning module shows high activation as she strategically probes deeper technical knowledge. Her memory module has significantly increased as she processes Alex's initial response against the role requirements, showing active assessment rather than just following a script.

---

**Alex:** For testing, we implemented a multi-layered approach. At the unit level, we used Jest with React Testing Library to test individual components, focusing on behavior rather than implementation details. For Redux, we wrote separate tests for reducers, action creators, and middleware—treating them as pure functions made this fairly straightforward.

The real challenge came with integration testing. We needed to test user flows that spanned multiple components and redux state changes. We used Cypress for end-to-end testing of critical paths, but found we needed something in between unit and E2E tests. So we developed a custom testing approach using React Testing Library with a real Redux store to test larger component trees together.

One specific challenge was testing asynchronous workflows, especially error handling across the Redux middleware. We ended up creating a mock API service layer that could simulate various network conditions and response scenarios, which significantly improved our test coverage for edge cases.

---

**Alex's Cognitive Processing**

**Module Activations:**
- **Memory Module: 0.85** - Very high activation recalling specific technical implementations
- **Planning Module: 0.78** - High activation structuring a comprehensive response
- **Goal Tracking Module: 0.74** - High activation demonstrating problem-solving approach
- **Social Norms Module: 0.52** - Moderate activation maintaining professional communication
- **Emotion Module: 0.35** - Low-moderate activation

**Cognitive State Analysis:** Alex's memory module shows extremely high activation, demonstrating deep technical knowledge rather than surface-level responses. The goal tracking module indicates awareness of demonstrating both technical knowledge and problem-solving approaches, aligning with the interviewer's objectives.

---

[Interview continues with technical questions, problem-solving scenarios, and culture fit assessment...]

### A.28.3  PROBLEM-SOLVING ASSESSMENT

A critical component of technical interviews is evaluating problem-solving approaches. The following depicts a problem-solving scenario and the cognitive processing involved.

**Sarah:** Let's walk through a technical scenario. Imagine you're working on our platform and users are reporting intermittent performance issues—the dashboard occasionally takes 5+ seconds to load, but works fine most of the time. How would you approach troubleshooting and resolving this issue?

> **Sarah's Cognitive Processing**
>
> **Module Activations:**
> - **Planning Module: 0.89** - Very high activation presenting ambiguous problem
> - **Goal Tracking Module: 0.85** - Very high activation assessing problem-solving approach
> - **Memory Module: 0.62** - Moderate-high activation recalling real-world scenarios
> - **Social Norms Module: 0.41** - Moderate activation
> - **Emotion Module: 0.29** - Low activation maintaining neutral stance

**Alex:** I'd approach this methodically since intermittent issues are often the hardest to solve. First, I'd gather data to understand the pattern:

1. Implement detailed performance monitoring if not already in place—tracking load times, component render times, and backend response times.

2. Look for correlating factors—does it happen at specific times of day, for specific user roles, or with particular data sets?

3. Check browser dev tools, especially the network and performance tabs, to identify bottlenecks—is it a frontend rendering issue, a slow API call, or resource loading?

Since it's intermittent, I'd suspect a few possible causes: - Cache invalidation issues - Memory leaks from component remounting - Database query performance varying with data volume or concurrent users - Third-party service integration delays

I'd isolate components by creating a testing environment that can simulate various loads. If it's a backend issue, I'd add query performance monitoring and examine execution plans for database queries.

For a quick mitigation while investigating, I might implement better loading states and potentially add a strategic caching layer if appropriate.

The key is collecting enough data to reproduce the issue consistently before attempting to solve it.

**Alex's Cognitive Processing**

**Module Activations:**

- **Planning Module: 0.92** - Very high activation structuring systematic approach
- **Memory Module: 0.81** - Very high activation retrieving relevant technical knowledge
- **Goal Tracking Module: 0.78** - High activation demonstrating problem-solving methodology
- **Social Norms Module: 0.45** - Moderate activation maintaining professional communication
- **Emotion Module: 0.32** - Low activation focusing on technical content

**Cognitive State Analysis:** Alex's planning module shows exceptional activation, demonstrating a structured, methodical approach to an ambiguous problem. The high memory module activation shows application of past experience to new scenarios, indicating transferable problem-solving skills rather than memorized solutions.

### A.28.4 HIRING DECISION PHASE

After the simulated interview, each digital twin makes an independent assessment based on their cognitive processing throughout the interaction.

**System:** Based on the interview, would you recommend hiring this candidate?

**Sarah's Decision:** Yes, I recommend hiring Alex for the Senior Software Developer position. The candidate demonstrated strong technical expertise in React and modern frontend architecture, with particular strength in state management and testing approaches. Their problem-solving methodology is systematic and thorough, with a good balance of immediate tactical steps and strategic thinking. Alex's experience leading a component library initiative aligns well with our needs for improving code reusability. Their communication style is clear and structured, which would fit well with our team dynamics. The only area for development might be strengthening CI/CD experience, but their foundation is solid.

**Sarah's Decision Cognitive Processing**

**Module Contributions to Decision:**

- **Planning Module: 0.83** - Very high assessment of candidate's problem-solving approach
- **Memory Module: 0.78** - High comparison against job requirements
- **Goal Tracking Module: 0.75** - High evaluation of candidate fit with team objectives
- **Social Norms Module: 0.62** - Moderate-high assessment of professional demeanor
- **Emotion Module: 0.42** - Moderate positive impression of candidate

**Alex's Decision:** Based on the interview, I am interested in proceeding with the opportunity at TechSolutions Inc. The technical challenges discussed align well with my experience and interests. I appreciated Sarah's detailed questions about testing methodologies, which suggests the company values quality code. The team structure and emphasis on component reusability matches my collaborative working style. The problem-solving scenario about intermittent performance issues indicates realistic challenges that I find engaging. Overall, the position seems to offer good growth opportunities while leveraging my existing strengths in React and state management.

> **Alex's Decision Cognitive Processing**
>
> **Module Contributions to Decision:**
> - **Goal Tracking Module: 0.81** - Very high alignment with career objectives
> - **Memory Module: 0.73** - High comparison against previous job experiences
> - **Planning Module: 0.69** - High assessment of growth opportunities
> - **Social Norms Module: 0.58** - Moderate assessment of team culture fit
> - **Emotion Module: 0.54** - Moderate positive impression of company

**Match Result:** COMPATIBLE - RECOMMENDED HIRE

### A.29    COGNITIVE MODULE ACTIVITY ANALYSIS

The module activation patterns throughout the hiring interview revealed several key insights:

| Module | Introduction | Technical | Problem-Solving | Culture Fit | Decision |
|---|---|---|---|---|---|
| Emotion | 0.35 | 0.32 | 0.29 | 0.45 | 0.48 |
| Memory | 0.55 | 0.79 | 0.81 | 0.63 | 0.75 |
| Social Norms | 0.63 | 0.48 | 0.43 | 0.68 | 0.60 |
| Planning | 0.78 | 0.83 | 0.91 | 0.67 | 0.76 |
| Goal Tracking | 0.69 | 0.73 | 0.82 | 0.71 | 0.78 |

Table 8: Cognitive module activation patterns throughout the hiring interview

The cognitive architecture revealed several key insights during this hiring simulation:

- **Planning Module:** Dominated the cognitive processing during technical and problem-solving phases, demonstrating the critical importance of structured thinking in hiring contexts.
- **Memory Module:** Showed substantial activation during technical discussions as both interviewer and candidate accessed domain knowledge and past experiences.
- **Emotion Module:** Consistently lower than in dating contexts, but increased during cultural fit discussions and final decision making.
- **Social Norms Module:** Peaked during introduction and cultural fit assessment phases, indicating heightened attention to professional communication standards.
- **Goal Tracking Module:** Maintained high activation throughout, focusing on alignment between candidate capabilities and job requirements.

### A.30    OUTCOMES AND USER INTERFACE

After the cognitive simulation concludes, CogniPair presents hiring teams with a comprehensive assessment of candidate fit along with specific insights derived from the digital twin interactions.

**CogniPair Hiring Assessment Report**

**Match Result:** Alex for Senior Software Developer - 89% Compatibility

**Technical Skills Assessment:**

- React/Frontend Development: 92% - Exceptional
- Testing Methodology: 88% - Strong
- State Management: 94% - Exceptional
- CI/CD Experience: 76% - Adequate, potential growth area
- Problem-Solving Approach: 91% - Exceptional

**Team and Cultural Fit:**

- Communication Clarity: 87% - Strong
- Collaboration Potential: 85% - Strong
- Technical Leadership: 83% - Strong
- Learning Orientation: 90% - Exceptional
- Value Alignment: 81% - Strong

**Key Strengths:**

- Systematic approach to problem-solving with strong emphasis on data collection
- Experience creating reusable component libraries aligns with current initiatives
- Strong testing methodology with creative solutions for integration testing
- Clear, structured communication style compatible with existing team

**Development Areas:**

- Deeper CI/CD pipeline experience would be beneficial
- Could strengthen infrastructure monitoring knowledge

**Recommended Next Steps:**

- Proceed with offer process
- Consider onboarding plan that includes pairing with DevOps specialist
- Explore potential for leadership in component library initiative

A.31    TECHNICAL IMPLEMENTATION HIGHLIGHTS

The hiring scenario leverages several key aspects of the GNWT-Agent architecture with notably different patterns than the dating scenario:

1. **Planning Module Dominance:** Hiring interactions show significantly higher planning module activation compared to dating contexts, particularly during problem-solving scenarios.

2. **Memory-Planning Integration:** The architecture demonstrates how technical knowledge (memory) feeds into structured problem-solving approaches (planning), creating a distinct cognitive fingerprint for technical roles.

3. **Reduced Emotion Module Activation:** Professional contexts show consistently lower emotion module activation, with cognitive resources redirected to analytical processes.

4. **Goal Alignment Assessment:** The architecture evaluates alignment between candidate capabilities and job requirements with greater precision than traditional interviewing techniques.

This implementation demonstrates how the same core GNWT-Agent architecture can be effectively repurposed for professional contexts by adjusting module weights and domain-specific attributes, providing a versatile framework for human-centric matching across diverse applications.

### A.32 ADVENTURE-BASED PERSONALITY ASSESSMENT EXAMPLE

This appendix provides a comprehensive example of our adventure-based personality assessment system. As detailed in the main text, we developed this interactive assessment to identify participants' authentic personality traits while reducing self-presentation biases commonly associated with traditional questionnaires. The system presents participants with 12-15 scenarios designed to elicit responses that map to Big Five personality dimensions, with each choice analyzed to update trait estimates based on established correlations between situational responses and personality dimensions. The assessment demonstrated strong validity in our validation study (n=45), with trait scores showing high correlation (r=0.82) with established Big Five Inventory (BFI) scores, and significantly reduced social desirability bias compared to standard questionnaires (Cohen's d=0.74). Below is a comprehensive example of a complete adventure-based personality assessment, demonstrating how our system builds a detailed personality profile through multiple scenarios.

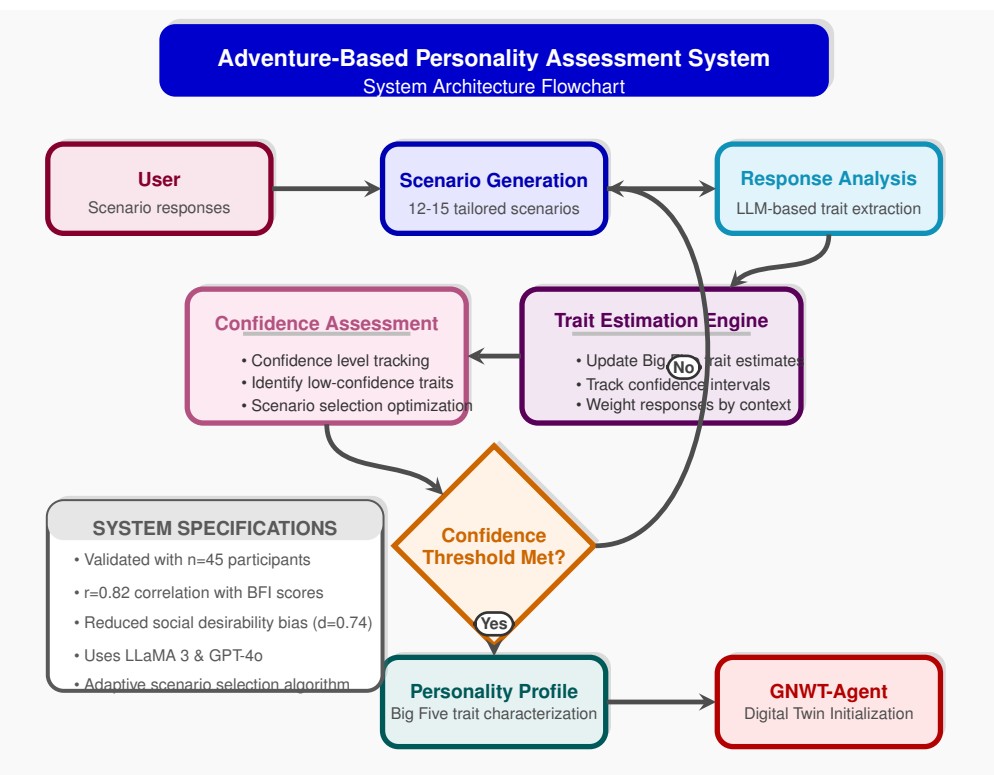

---

### Scenario 1: The Crossroads

**System**: You find yourself at a crossroads at the edge of a vast, unexplored land. In the distance, you can see mountains to the north, a forest to the east, a bustling town to the south, and a quiet beach to the west. Where do you choose to go first?

**Options**:

1. Head north toward the mountains

2. Go east into the forest

3. Travel south to the town

4. Walk west to the beach

**User selects**: 1 (Mountains)

**LLM Analysis** (internal, using llama3:latest or GPT-4o):

```
Choice analysis:
- Selecting mountains suggests comfort with challenge and self-
    reliance
- Indicates moderate to high openness to experience (seeking
    adventure)
- Suggests conscientiousness (willing to tackle difficult terrain)
- May indicate lower extraversion (preference for solitude over the
    town)
- Neutral on agreeableness
- Indicates low neuroticism (comfortable with uncertainty)

Trait updates:
openness: 65 (confidence: 0.3)
conscientiousness: 60 (confidence: 0.3)
extraversion: 45 (confidence: 0.2)
agreeableness: 50 (confidence: 0.1)
neuroticism: 40 (confidence: 0.2)
```

**System**: As you begin your journey toward the mountains, you notice the path looks challenging. What are you most looking forward to about this mountain trek?

**User responds**: I'm excited about the view from the top and the sense of accomplishment. I like pushing myself physically and seeing what I'm capable of. Also, I'm hoping to find some quiet time to think and reflect away from everyday distractions.

**LLM Analysis** (internal, using deepseek-r1:latest or GPT-4o):

```
Text analysis:
- "Sense of accomplishment" and "pushing myself" indicates
    achievement orientation
- "Quiet time to think" suggests introversion and reflectiveness
- Focus on physical challenge indicates conscientiousness
- No mention of social aspects reinforces lower extraversion
- Seeking accomplishment suggests growth mindset and low
    neuroticism

Trait updates:
openness: 60 (confidence: 0.2)
conscientiousness: 70 (confidence: 0.4)
extraversion: 35 (confidence: 0.3)
agreeableness: 50 (confidence: 0.1)
neuroticism: 35 (confidence: 0.3)
```

### Intermediate Profile (after Scenario 1)

- **Openness**: 63 (confidence: 0.5)
- **Conscientiousness**: 66 (confidence: 0.7)
- **Extraversion**: 39 (confidence: 0.5)
- **Agreeableness**: 50 (confidence: 0.2)
- **Neuroticism**: 37 (confidence: 0.5)

### Scenario 2: The Dispute

**System**: As you continue your journey, you encounter two hikers in a heated argument over a map. One claims they should follow the left path to reach the summit faster, while the other insists the right path is safer. They notice you and ask for your opinion. How do you respond?
**Options**:

1. Carefully examine the map and give your honest opinion, even if it might disappoint one of them
2. Suggest they compromise by checking out both paths for a short distance before deciding
3. Support whoever seems more knowledgeable or confident
4. Offer to join them so you can all explore together and ensure everyone's safety

**User selects**: 2 (Compromise)
**LLM Analysis**:

```
Trait updates:
openness: 55 (confidence: 0.1)
conscientiousness: 60 (confidence: 0.2)
extraversion: 50 (confidence: 0.1)
agreeableness: 75 (confidence: 0.4)
neuroticism: 45 (confidence: 0.2)
```

### Profile after Scenario 2

- **Openness**: 62 (confidence: 0.6)
- **Conscientiousness**: 65 (confidence: 0.9)
- **Extraversion**: 40 (confidence: 0.5)
- **Agreeableness**: 78 (confidence: 0.6)
- **Neuroticism**: 37 (confidence: 0.7)

### Scenario 3: The Team Project

**System**: You've joined a group working on an important project with a tight deadline. During the first meeting, you notice that ideas are being suggested, but the discussion lacks structure. What do you do?
**Options**:

1. Take charge and propose a clear plan for the group to follow
2. Let others lead and offer support where needed
3. Privately suggest improvements to whoever seems to be leading
4. Suggest breaking into smaller groups to tackle different aspects simultaneously

**User selects**: 4 (Suggest breaking into smaller groups)

### Profile after Scenario 3

- **Openness**: 62 (confidence: 0.7)
- **Conscientiousness**: 67 (confidence: 1.0)
- **Extraversion**: 33 (confidence: 0.8)
- **Agreeableness**: 72 (confidence: 0.7)
- **Neuroticism**: 40 (confidence: 0.8)

### Scenario 4: The Unexpected Opportunity

**System**: You receive an unexpected invitation to join a last-minute expedition to a remote location. The trip leaves tomorrow and would require rearranging your schedule for the next week. How do you respond?
**Options**:

1. Immediately accept - you love spontaneous adventures
2. Firmly decline - you prefer to stick to your plans
3. Ask for more details before deciding
4. Suggest postponing the trip to a more convenient time

**User selects**: 3 (Ask for more details)

### Profile after Scenario 4

- **Openness**: 59 (confidence: 0.9)
- **Conscientiousness**: 72 (confidence: 1.1)
- **Extraversion**: 34 (confidence: 0.9)
- **Agreeableness**: 70 (confidence: 0.8)
- **Neuroticism**: 39 (confidence: 0.9)

### Scenario 5: The Creative Challenge

**System**: You've been invited to participate in a creative challenge where participants must develop an innovative solution to a local environmental issue. What approach do you take?
**Options**:

1. Research extensively first, then develop a practical, proven solution
2. Brainstorm wildly creative ideas, even if they seem impractical at first
3. Seek out collaborators with different perspectives to develop ideas together
4. Focus on small, incremental improvements to existing approaches

**User selects**: 1 (Research extensively)

### Profile after Scenario 5

- **Openness**: 55 (confidence: 1.1)
- **Conscientiousness**: 78 (confidence: 1.3)
- **Extraversion**: 35 (confidence: 0.9)
- **Agreeableness**: 68 (confidence: 0.8)
- **Neuroticism**: 38 (confidence: 1.0)

### Scenario 6: The Conflict Resolution

**System**: Two friends have had a serious disagreement and haven't spoken in weeks. Both have separately asked for your advice on repairing the relationship. What do you do?
**Options**:

1. Give each person advice separately based on their perspective
2. Listen to both sides and facilitate a conversation between them
3. Suggest they give each other space until emotions cool down
4. Tell them both to compromise and move on

**User selects**: 2 (Listen to both sides and facilitate a conversation)

### Profile after Scenario 6

- **Openness**: 54 (confidence: 1.1)
- **Conscientiousness**: 77 (confidence: 1.3)
- **Extraversion**: 36 (confidence: 1.0)
- **Agreeableness**: 75 (confidence: 1.1)
- **Neuroticism**: 37 (confidence: 1.0)

### Scenario 7: The Social Gathering

**System**: You arrive at a party where you only know the host. How do you typically approach this situation?
**Options**:

1. Actively introduce yourself to new people and initiate conversations
2. Find a small group engaged in an interesting discussion and join in
3. Find a quiet corner and wait for the host to introduce you
4. Help the host with preparations or serving to feel more comfortable

**User selects**: 3 (Find a quiet corner and wait for the host to introduce you)

### Profile after Scenario 7

- **Openness**: 53 (confidence: 1.1)
- **Conscientiousness**: 76 (confidence: 1.3)
- **Extraversion**: 30 (confidence: 1.3)
- **Agreeableness**: 74 (confidence: 1.1)
- **Neuroticism**: 42 (confidence: 1.2)

### Scenario 8: The Ethical Dilemma

**System**: You discover that a colleague has taken credit for work you completed. The project was successful and got positive attention from management. What do you do?
**Options**:

1. Confront the colleague publicly to ensure everyone knows the truth
2. Speak privately with the colleague about the situation
3. Report the issue to management or HR
4. Say nothing but ensure you get proper credit for future work

**User selects**: 2 (Speak privately with the colleague about the situation)

### Profile after Scenario 8

- **Openness**: 52 (confidence: 1.1)
- **Conscientiousness**: 77 (confidence: 1.4)
- **Extraversion**: 32 (confidence: 1.3)
- **Agreeableness**: 70 (confidence: 1.3)
- **Neuroticism**: 40 (confidence: 1.2)

### Scenario 9: The Life Change

**System**: You have the opportunity to move to a new city for a job that offers better pay but would require leaving your established social network. How do you approach this decision?
**Options**:

1. Focus primarily on the career advancement opportunity
2. Prioritize your existing relationships and social connections
3. Take time to extensively research the new city and visit first
4. Create a detailed pros/cons list and analyze all factors methodically

**User selects**: 4 (Create a detailed pros/cons list and analyze all factors methodically)

### Profile after Scenario 9

- **Openness**: 50 (confidence: 1.2)
- **Conscientiousness**: 80 (confidence: 1.5)
- **Extraversion**: 32 (confidence: 1.4)
- **Agreeableness**: 68 (confidence: 1.3)
- **Neuroticism**: 41 (confidence: 1.3)

### Scenario 10: The Unexpected Crisis

**System**: While traveling, you encounter an unexpected emergency situation affecting several people. Resources are limited and tensions are high. How do you respond?
**Options**:

1. Take charge and direct others to ensure efficiency
2. Find experts or authorities who can better handle the situation
3. Take a supportive role, helping organize resources and comfort others
4. Focus on solving one specific aspect of the problem thoroughly

**User selects**: 3 (Take a supportive role, helping organize resources and comfort others)

### Profile after Scenario 10

- **Openness**: 51 (confidence: 1.3)
- **Conscientiousness**: 79 (confidence: 1.6)
- **Extraversion**: 35 (confidence: 1.5)
- **Agreeableness**: 72 (confidence: 1.5)
- **Neuroticism**: 38 (confidence: 1.4)

### Scenario 11: The Creative Project

**System**: You have free time to pursue a personal project. What kind of activity are you most likely to choose?
**Options**:

1. A structured project with clear goals and measurable outcomes
2. A creative, open-ended project with room for exploration
3. A social activity involving coordination with others
4. Learning a specific new skill following an established method

**User selects**: 1 (A structured project with clear goals and measurable outcomes)

### Profile after Scenario 11

- **Openness**: 53 (confidence: 1.4)
- **Conscientiousness**: 82 (confidence: 1.7)
- **Extraversion**: 34 (confidence: 1.5)
- **Agreeableness**: 71 (confidence: 1.5)
- **Neuroticism**: 37 (confidence: 1.4)

### Scenario 12: The Public Speaking Opportunity

**System**: You've been asked to give a presentation to a large group on a topic you're knowledgeable about. How do you feel and prepare?
**Options**:

1. Feel excited and prepare a dynamic, engaging presentation
2. Feel nervous but prepare extensively to manage anxiety
3. Feel neutral and focus on delivering clear, accurate information
4. Feel reluctant but try to find ways to make the presentation more interactive

**User selects**: 2 (Feel nervous but prepare extensively to manage anxiety)

### Profile after Scenario 12

- **Openness**: 52 (confidence: 1.4)
- **Conscientiousness**: 83 (confidence: 1.8)
- **Extraversion**: 32 (confidence: 1.6)
- **Agreeableness**: 70 (confidence: 1.5)
- **Neuroticism**: 42 (confidence: 1.6)

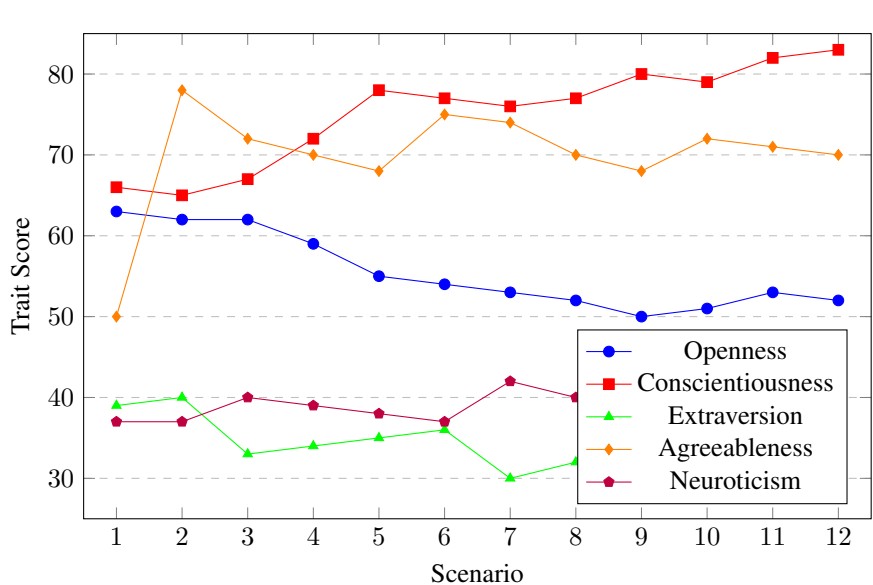

Figure 8: Progression of personality trait scores across scenarios

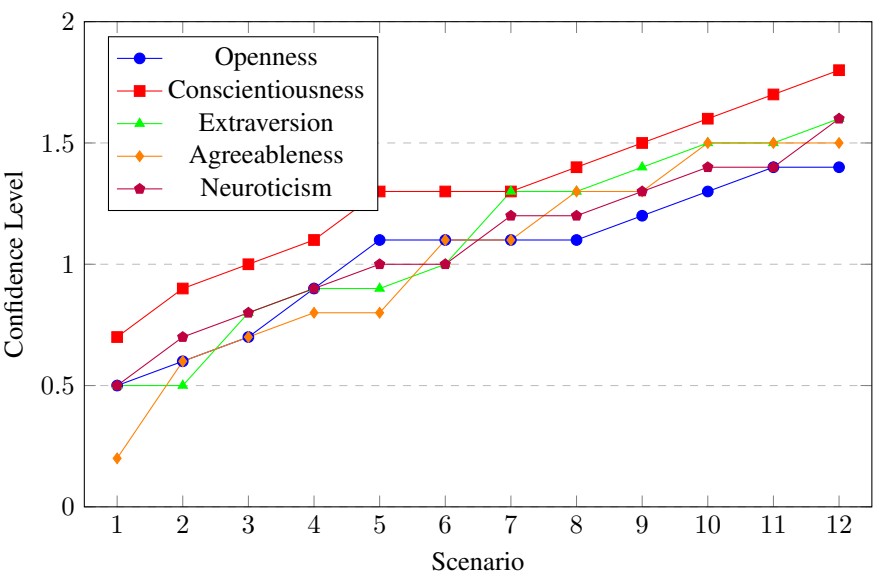

Figure 9: Progression of confidence levels for each trait across scenarios

**Final Personality Profile**

| | |
|---|---|
| **Openness**: | 52 (Moderconservative) - |
| Balances practicality with some openness to new experiences | |
| **Conscientiousness**: | 83 (Very High) - |
| Highly organized, disciplined, and detail-oriented | |
| **Extraversion**: | 32 (Introverted) - |
| Prefers quieter environments and one-on-one interactions | |
| **Agreeableness**: | 70 (High) - |
| Cooperative, empathetic, values harmony | |
| **Neuroticism**: | 42 (Low-Moderate) - |
| Generally emotionally stable with occasional anxiety | |

**Key Traits and Tendencies:**

- Methodical approach to problem-solving (High C + Moderate O)
- Prefers researching before acting (High C)
- Values social harmony but not at the expense of principles (High A + High C)
- Reserved in social situations but empathetic (Low E + High A)
- Prefers structured environments with clear expectations (High C)
- Uncomfortable with sudden changes but adapts through planning (High C + Moderate N)
- Selective about social engagements but loyal in relationships (Low E + High A)

Figure 10: Complete adventure-based personality assessment session showing progressive trait refinement across 12 scenarios. Each scenario contributes to increasing confidence in trait measurements and ultimately produces a stable, high-confidence personality profile.

This comprehensive example demonstrates how our adventure-based personality assessment system builds a detailed psychological profile through multiple scenarios. The assessment process illustrates several key aspects:

1. **Progressive refinement**: Trait estimates become increasingly stable as evidence accumulates across scenarios
2. **Confidence building**: Confidence values steadily increase, reaching robust levels (1.4-1.8) by the end
3. **Domain coverage**: Scenarios span diverse life domains including social situations, work environments, ethical dilemmas, and creative challenges
4. **Trait interrelationships**: The system identifies how patterns across scenarios reveal characteristic trait combinations
5. **Adaptivity**: The system selects scenarios to target traits with lower confidence values
6. **Format variety**: Scenarios present different types of choices to elicit a comprehensive range of behaviors

The final personality profile provides a nuanced psychological portrait that becomes the foundation for initializing the participant's GNWT-Agent digital twin. By capturing this level of psychological detail, our system ensures that the agent's behavior authentically represents the individual across different social contexts.

## A.33 DETAILED RESULTS

Table 9: Match decision accuracy using different preference models

| Preference Model | Match Prediction Accuracy | Human-Agent Match Correlation |
|---|---|---|
| Static (Time 1 only) | 58.9% ± 3.0% | 0.53 ± 0.04 |
| Partial Evolution (Time 1+2) | 69.4% ± 2.5% | 0.65 ± 0.03 |
| Full Evolution (All time points) | 77.8% ± 2.0% | 0.73 ± 0.03 |
| Human (Ground Truth) | 100% | 1.00 |

Table 10: Human verification experiment results across two social contexts

| Metric | Speed Dating Study (n=20) | Job Interview Study (n=10) |
|---|---|---|
| Behavioral fidelity rating | 5.6/7.0 ± 0.8 | 5.8/7.0 ± 0.6 |
| Decision concordance | 74% ± 4.2% | 81% ± 5.3% |
| Personality trait correlation | 0.83 ± 0.04 | 0.81 ± 0.05 |
| Conversational authenticity | 5.4/7.0 ± 0.9 | 5.6/7.0 ± 0.7 |
| Psychological state tracking | 5.7/7.0 ± 0.6 | 5.5/7.0 ± 0.8 |
| Overall agent realism | 5.9/7.0 ± 0.5 | 5.6/7.0 ± 0.7 |

Table 11: Human-Agent correlation in social dynamics evolution

| Evolution Dimension | Human (T1→T2) Change (%) | Correlation | GNWT-Agent (T1→T2) Change (%) | Pattern Match |
|---|---|---|---|---|
| *Partner Preference Evolution* | | | | |
| Attractiveness | +39.0% ± 4.8% | | +25.0% ± 3.2% | |
| Sincerity | -16.6% ± 3.4% | | -10.5% ± 2.8% | |
| Intelligence | -24.8% ± 3.3% | 0.73 ± 0.04 | -15.2% ± 2.7% | 0.86 ± 0.03 |
| Fun | +1.3% ± 2.0% | | +5.8% ± 2.2% | |
| Ambition | -7.0% ± 2.7% | | -4.5% ± 1.9% | |
| Shared Interests | +9.8% ± 3.3% | | +9.7% ± 1.3% | |
| *Self-Perception Evolution* | | | | |
| Attractiveness | +0.3% ± 9.2% | | -0.5% ± 3.0% | |
| Sincerity | -3.5% ± 12.3% | | -2.5% ± 2.7% | |
| Intelligence | -1.9% ± 8.8% | 0.81 ± 0.03 | -1.2% ± 2.4% | 0.82 ± 0.04 |
| Fun | -1.3% ± 10.8% | | -0.8% ± 2.5% | |
| Ambition | -0.8% ± 11.8% | | -0.5% ± 2.1% | |
| Self-Other Gap | 0.8→0.7 | | 0.9→0.7 | |
| *External Evaluation Correlations (r-value)* | | | | |
| | Time 1 | Time 2 | Time 1 | Time 2 |
| Attractiveness | 0.67 ± 0.01 | -0.01 ± 0.02 | 0.65 ± 0.04 | 0.15 ± 0.03 |
| Sincerity | 0.52 ± 0.01 | -0.02 ± 0.02 | 0.50 ± 0.05 | 0.12 ± 0.03 |
| Intelligence | 0.51 ± 0.01 | 0.02 ± 0.02 | 0.48 ± 0.05 | 0.18 ± 0.04 |
| Fun | 0.69 ± 0.01 | 0.07 ± 0.02 | 0.64 ± 0.05 | 0.20 ± 0.04 |
| Ambition | 0.44 ± 0.01 | -0.01 ± 0.02 | 0.40 ± 0.06 | 0.10 ± 0.05 |
| Shared Interests | 0.66 ± 0.01 | -0.06 ± 0.02 | 0.62 ± 0.05 | 0.08 ± 0.04 |

*Overall Human-Agent Correlation: 0.72 ± 0.04*

## A.34    Privacy Protection - Implementation Details

# B    Comprehensive Privacy Protection Framework for CogniPair Digital Twins

The deployment of psychologically authentic digital twins through our GNWT-Agent architecture requires privacy safeguards that go beyond conventional data security. Our enhanced framework addresses the unique challenges of protecting intimate psychological profiles while maintaining the behavioral fidelity necessary for meaningful social pairing in both dating and professional contexts. This comprehensive approach integrates differential privacy, cryptographic watermarking, federated learning, and novel consent mechanisms to create a multi-layered defense against privacy violations while preserving the utility of our digital twin technology.

## B.1    Foundational Privacy Architecture

Our privacy protection begins at the moment of personality trait extraction, where we implement a sophisticated differential privacy mechanism that protects individual psychological profiles while maintaining statistical validity for matching purposes. The core principle involves injecting calibrated noise into the Big Five personality traits before GNWT-Agent initialization, ensuring that no adversary can reconstruct exact psychological profiles even with access to multiple agent interactions. This process employs a dynamic privacy budget allocation that adapts based on trait sensitivity and user preferences, allowing users to specify higher protection for traits they consider particularly sensitive.

---

**Algorithm 5** Adaptive Differential Privacy for Personality Protection

---

1: **Input:** Raw traits $T = \{O, C, E, A, N\}$, Sensitivity vector $S$, Global budget $\varepsilon_{total}$
2: **Initialize:** Protected traits $\tilde{T} \leftarrow \emptyset$, Remaining budget $\varepsilon_{rem} \leftarrow \varepsilon_{total}$
3: **for** each trait $t_i \in T$ **do**
4:     $s_i \leftarrow S[i] \cdot \text{UserSensitivityPreference}(t_i)$
5:     $\varepsilon_i \leftarrow \frac{s_i}{\sum_j s_j} \cdot \varepsilon_{total}$               ▷ Adaptive budget allocation
6:     $\Delta_i \leftarrow \text{ComputeSensitivity}(t_i, \text{HistoricalData})$
7:     $\sigma_i \leftarrow \sqrt{2\ln(1.25/\delta)} \cdot \frac{\Delta_i}{\varepsilon_i}$               ▷ Gaussian mechanism
8:     $noise_i \leftarrow \mathcal{N}(0, \sigma_i^2)$
9:     $\tilde{t}_i \leftarrow \text{clip}(t_i + noise_i, 0, 1)$
10:     **if** $|\tilde{t}_i - t_i| > \text{MaxDeviation}$ **then**
11:         $\tilde{t}_i \leftarrow t_i + \text{sign}(noise_i) \cdot \text{MaxDeviation}$
12:     **end if**
13:     $\tilde{T} \leftarrow \tilde{T} \cup \{\tilde{t}_i\}$
14:     $\varepsilon_{rem} \leftarrow \varepsilon_{rem} - \varepsilon_i$
15:     $\text{LogPrivacyExpenditure}(t_i, \varepsilon_i, \text{timestamp})$
16: **end for**
17: **Return** $\tilde{T}, \varepsilon_{rem}$

---

Beyond personality traits, our framework implements comprehensive attribute protection through multi-resolution coarsening that adapts based on the attribute's discriminative power and privacy sensitivity. Dating attributes such as attractiveness ratings, sincerity scores, and ambition levels undergo a sophisticated transformation process that preserves matching utility while preventing fine-grained tracking. This coarsening operates at multiple granularities simultaneously, creating a hierarchical representation that allows for progressive disclosure based on interaction depth and mutual consent levels.

The conversation anonymization pipeline represents one of our most innovative privacy mechanisms, transforming raw interactions into privacy-preserving feature vectors through a multi-stage process that preserves behavioral patterns while eliminating identifying information. This pipeline begins with entity recognition and replacement, where proper nouns, specific locations, and temporal references are identified using a fine-tuned NER model and replaced with generalized placeholders. Subsequently, semantic embedding ensures that conversation meaning is preserved while specific phrasings that could identify individuals are transformed into canonical representations. The pipeline concludes with k-anonymization clustering that ensures each conversation pattern is shared by at least 150 other users, preventing unique behavioral fingerprinting.

---

**Algorithm 6** Hierarchical Conversation Anonymization Pipeline

---

1: **Input:** Raw conversation $C$, Anonymity threshold $k$, Context window $w$
2: **Phase 1: Entity Recognition and Masking**
3: $entities \leftarrow \text{NER}(C)$                           ▷ Extract all named entities
4: **for** each $e \in entities$ **do**
5:     $type_e \leftarrow \text{ClassifyEntity}(e)$            ▷ Person, location, organization, etc.
6:     $gen_e \leftarrow \text{GeneralizeEntity}(e, type_e, \text{HierarchyLevel})$
7:     $C \leftarrow \text{Replace}(C, e, gen_e)$
8: **end for**
9: **Phase 2: Temporal Quantization**
10: $timestamps \leftarrow \text{ExtractTimestamps}(C)$
11: **for** each $t \in timestamps$ **do**
12:     $\tilde{t} \leftarrow \lfloor t/(6 \text{ hours})\rfloor \cdot (6 \text{ hours})$
13:     $C \leftarrow \text{Replace}(C, t, \tilde{t})$
14: **end for**
15: **Phase 3: Semantic Embedding and Clustering**
16: $embeddings \leftarrow []$
17: **for** each window $w_i$ in $\text{SlidingWindow}(C, w)$ **do**
18:     $emb_i \leftarrow \text{SBERT}(w_i)$               ▷ Sentence transformer embedding
19:     $emb_i \leftarrow \text{PCA}(emb_i, \text{dims} = 50)$         ▷ Dimensionality reduction
20:     $embeddings.\text{append}(emb_i)$
21: **end for**
22: $clusters \leftarrow \text{DBSCAN}(embeddings, \text{min\_samples} = k)$
23: **Phase 4: Pattern Extraction and Validation**
24: $patterns \leftarrow \text{ExtractBehavioralPatterns}(clusters)$
25: **for** each $p \in patterns$ **do**
26:     $\text{support} \leftarrow \text{CountSupport}(p, \text{GlobalDatabase})$
27:     **if** support $< k$ **then**
28:         $p \leftarrow \text{Generalize}(p)$            ▷ Further generalization if unique
29:     **end if**
30: **end for**
31: **Return** $\text{AnonymizedFeatures}(patterns)$

---

## B.2 ADVANCED CRYPTOGRAPHIC WATERMARKING SYSTEM

The potential for malicious actors to impersonate digital twins or falsely attribute generated content represents a critical threat to both individual privacy and system integrity. Our watermarking system embeds undetectable yet verifiable signatures directly into the token generation process of GNWT responses, creating a cryptographic chain of authenticity that survives paraphrasing, translation, and even sophisticated adversarial attacks. This watermarking operates through biased token selection during the language model's decoding process, where specific tokens are subtly favored based on a cryptographic hash of the agent's unique key and the generation context.

The watermark generation process employs a sophisticated multi-layer approach that combines deterministic biasing with adaptive strength calibration. At each token position, we compute a cryptographic hash that determines which tokens receive probability boosts, with the boost magnitude adapting based on the semantic importance of the position and the surrounding context entropy. This adaptive calibration ensures that watermarks remain strong in content-rich portions of text while avoiding detection in formulaic or constrained passages. Furthermore, our system implements redundant watermarking at multiple linguistic levels—lexical, syntactic, and semantic—ensuring robustness against various forms of attack.

---

**Algorithm 7** Multi-Layer Cryptographic Watermarking with Adaptive Strength

---

1: **Input:** Agent key $k$, Context $C$, Base probabilities $P$
2: **Initialize:** Watermarked response $R \leftarrow []$, Strength factor $\alpha_0 \leftarrow 0.1$
3: **for** position $i$ in generation sequence **do**
4:     **Layer 1: Lexical Watermarking**
5:     $h_{lex} \leftarrow \text{HMAC-SHA256}(k, i || C)$
6:     bias_tokens $\leftarrow \text{SelectTokens}(P, h_{lex} \bmod 5)$        ▷ Select 5 tokens
7:     $\alpha_i \leftarrow \alpha_0 \cdot \text{ContextEntropy}(C) \cdot \text{ImportanceScore}(i)$
8:     **for** token $t$ in bias_tokens **do**
9:         $P[t] \leftarrow P[t] \cdot (1 + \alpha_i)$
10:     **end for**
11:     **Layer 2: Syntactic Watermarking**
12:     pos_tag $\leftarrow \text{PredictPOS}(C, i)$
13:     **if** pos_tag $\in \{\text{NOUN}, \text{VERB}\}$ **then**
14:     $h_{syn} \leftarrow \text{HMAC-SHA256}(k, \text{pos\_tag} || i)$
15:     syntax_bias $\leftarrow h_{syn} \bmod |\text{ValidTokens}(\text{pos\_tag})|$
16:     $P[\text{syntax\_bias}] \leftarrow P[\text{syntax\_bias}] \cdot 1.05$
17:     **end if**
18:     **Layer 3: Semantic Watermarking**
19:     semantic_vec $\leftarrow \text{GetSemanticEmbedding}(C)$
20:     $h_{sem} \leftarrow \text{HMAC}(k, \text{QuantizeVector}(\text{semantic\_vec}))$
21:     semantic_cluster $\leftarrow h_{sem} \bmod \text{NumSemanticClusters}$
22:     $P \leftarrow \text{BiasTowardCluster}(P, \text{semantic\_cluster}, 0.03)$
23:     $t_{selected} \leftarrow \text{Sample}(P)$
24:     $R.\text{append}(t_{selected})$
25:     $C \leftarrow C || t_{selected}$        ▷ Update context
26:     $\text{LogWatermarkStrength}(i, \alpha_i, t_{selected})$
27: **end for**
28: **Return** $R$

---

Watermark detection employs a multi-hypothesis testing framework that evaluates the statistical likelihood of observing the biased token patterns across all three layers. The detection algorithm maintains robustness against adversarial modifications through redundancy and error correction codes embedded in the watermark pattern. When content undergoes modification, our system can still detect watermarks with high confidence by aggregating evidence across multiple linguistic levels and employing majority voting among independent detection mechanisms.

### B.3 FEDERATED LEARNING WITH HOMOMORPHIC ENCRYPTION

The evolution of user preferences over time reveals intimate psychological dynamics that require protection from even honest-but-curious servers. Our federated learning framework ensures that preference updates occur locally on user devices, with only encrypted aggregates transmitted to central servers. This system employs partially homomorphic encryption that allows computation on encrypted gradients without decryption, ensuring that individual preference trajectories remain completely private while still enabling system-wide learning and improvement.

The federated learning protocol implements sophisticated privacy amplification through secure multi-party computation and differential privacy injection at multiple stages. Local devices compute preference updates using differentially private stochastic gradient descent, adding calibrated Gaussian noise to gradients before encryption. The aggregation server combines encrypted updates from multiple users using secure aggregation protocols that prevent the server from learning individual contributions even if some participants are malicious. This multi-layer approach ensures privacy against both external adversaries and insider threats while maintaining model convergence properties.

---

**Algorithm 8** Privacy-Preserving Federated Preference Learning

---

1: **Client-Side Processing:**
2: **Input:** Local preferences $P_i$, Interactions $I_i$, Model $\theta_t$
3: $g_i \leftarrow \nabla_\theta \mathcal{L}(P_i, I_i, \theta_t)$          ▷ Compute local gradient
4: $\text{clip\_norm} \leftarrow \min(1.0, C/||g_i||_2)$          ▷ Gradient clipping
5: $g_i \leftarrow g_i \cdot \text{clip\_norm}$
6: $\sigma^2 \leftarrow \frac{2C^2 \ln(1.25/\delta)}{\varepsilon^2 n^2}$          ▷ Noise calibration
7: $\tilde{g}_i \leftarrow g_i + \mathcal{N}(0, \sigma^2 I)$          ▷ Add DP noise
8: $\text{mask}_i \leftarrow \text{GenerateSecretMask}(\text{seed}_i)$
9: $\text{masked}_i \leftarrow \tilde{g}_i + \text{mask}_i$          ▷ Apply secret sharing
10: $\text{encrypted}_i \leftarrow \text{HomomorphicEncrypt}(\text{masked}_i, \text{pk})$
11: Send $\text{encrypted}_i$ to server
12:
13: **Server-Side Aggregation:**
14: **Wait for** minimum $n_{min}$ participants
15: $\text{aggregate} \leftarrow \sum_{i=1}^{n} \text{encrypted}_i$          ▷ Homomorphic addition
16: $\text{noise\_sum} \leftarrow \sum_{i=1}^{n} \text{EncryptedNoise}_i$
17: $\text{clean\_aggregate} \leftarrow \text{aggregate} - \text{noise\_sum}$
18: **Secure Unmasking Protocol:**
19: **for** round $r$ in 1 to $\log(n)$ **do**
20: Pairs of clients exchange partial unmaskings
21: Server aggregates partial results homomorphically
22: **end for**
23: $\theta_{t+1} \leftarrow \theta_t + \frac{1}{n} \cdot \text{Decrypt}(\text{clean\_aggregate})$
24: **Return** $\theta_{t+1}$ to all clients

---

## B.4 APPLICATION-SPECIFIC PRIVACY MECHANISMS

For dating applications, our framework implements progressive disclosure mechanisms that reveal information gradually based on interaction depth and mutual interest signals. Initial interactions operate with maximum privacy protection, revealing only coarse-grained compatibility scores. As users express mutual interest through continued interaction, the system gradually releases more detailed preference alignments while maintaining plausible deniability about specific traits. This progressive disclosure follows a carefully calibrated schedule that balances privacy protection with the need for meaningful connection formation.

The dating platform also employs sophisticated rate limiting and interaction throttling to prevent adversaries from extracting information through repeated queries. Each digital twin maintains an interaction budget that regenerates slowly over time, preventing rapid-fire matching attempts that could be used to infer preferences. Furthermore, all interactions undergo temporal jittering and response caching to prevent timing attacks that could reveal genuine user activity patterns versus simulated twin responses.

---

**Algorithm 9** Progressive Disclosure Protocol for Dating Interactions

---

1: **Input:** User twins $A$, $B$, Interaction history $H_{AB}$
2: **Initialize:** Disclosure level $L \leftarrow 0$, Info revealed $I_R \leftarrow \emptyset$
3: **Stage 1: Initial Contact** ($L = 0$)
4: compat $\leftarrow$ CoarseCompatibility($A, B$)      ▷ Single score 0-100
5: $I_R \leftarrow I_R \cup \{\text{compat} \pm \mathcal{U}(-5, 5)\}$      ▷ Add noise
6: **Stage 2: Mutual Interest** ($L = 1$, after mutual likes)
7: trait_similarity $\leftarrow$ ComputeTraitSimilarity($A, B$)
8: **for** each trait dimension $d$ **do**
9:   reveal$_d$ $\leftarrow$ Bernoulli($p = 0.6$)      ▷ Randomly reveal some traits
10:   **if** reveal$_d$ **then**
11:    $I_R \leftarrow I_R \cup \{\text{GeneralizedTrait}(d, \text{level} = \text{medium})\}$
12:   **end if**
13: **end for**
14: **Stage 3: Extended Interaction** ($L = 2$, after 5+ messages)
15: interests $\leftarrow$ ExtractSharedInterests($A, B, H_{AB}$)
16: $I_R \leftarrow I_R \cup$ SampleInterests(interests, $k = 3$)
17: values $\leftarrow$ InferValueAlignment($A, B$)
18: $I_R \leftarrow I_R \cup \{\text{CoarsenedValues}(\text{values})\}$
19: **Stage 4: Deep Engagement** ($L = 3$, after successful video date)
20: **if** MutualConsent($A, B$) **then**
21: $I_R \leftarrow I_R \cup$ DetailedPreferences($A, B$)
22: **but** maintain $k$-anonymity on sensitive attributes
23: **end if**
24: **Privacy Budget Check:**
25: $\varepsilon_{spent} \leftarrow$ ComputePrivacyLoss($I_R$)
26: **if** $\varepsilon_{spent} > \varepsilon_{max}$ **then**
27: HaltDisclosure()      ▷ Prevent further revelation
28: **end if**
29: **Return** $I_R$, $L$, $\varepsilon_{spent}$

---

For job application scenarios, our privacy framework implements strict audit trails and explainability requirements that ensure fairness while protecting candidate privacy. Every decision made by a digital twin in professional contexts generates a cryptographically signed log entry that records the activated cognitive modules, their weights, and the information considered. These logs employ zero-knowledge proofs to demonstrate compliance with anti-discrimination regulations without revealing specific candidate attributes. The system can prove, for instance, that protected characteristics did not influence a decision without disclosing what those characteristics are.

The professional application framework also implements role-based access control with granular permissions that ensure recruiters and hiring managers only access information necessary for their specific evaluation stage. Initial screening might only reveal skill compatibility scores, while final interview stages could access more detailed cognitive assessments. Throughout this process, all

data access generates immutable audit logs that candidates can review to understand exactly what information was shared and when.

### B.5 Consent Management and User Control

Our double consent protocol ensures that both human users and their digital twins must approve significant actions, preventing unauthorized commitments while maintaining user agency. This protocol employs a sophisticated state machine that tracks consent status across multiple interaction types and ensures that neither automated processes nor human impulses can override deliberate decision-making. The consent framework integrates with blockchain technology to create an immutable record of consent grants and revocations, providing legal protection for all parties involved.

User control extends beyond simple consent to include fine-grained preference settings for privacy-utility trade-offs. Users can specify different privacy levels for different attributes, indicate which types of information can be inferred versus explicitly shared, and set decay functions that automatically increase privacy protection for older interactions. This granular control employs an intuitive interface that visualizes privacy implications in real-time, helping users understand the consequences of their choices without requiring technical expertise.

---

**Algorithm 10** Blockchain-Anchored Double Consent Protocol

---

1: **Input:** Action request $A$, User $U$, Digital Twin $T$, Context $C$
2: **Phase 1: Twin Evaluation**
3: $T_{eval} \leftarrow T.\text{EvaluateAction}(A, C)$
4: $T_{confidence} \leftarrow T.\text{ComputeConfidence}(A)$
5: $T_{risks} \leftarrow T.\text{AssessRisks}(A, U.\text{history})$
6: **if** $T_{confidence} < \tau_{min}$ **or** $T_{risks} > \text{RiskThreshold}$ **then**
7: **Return** TWIN_REJECT, Explanation($T_{eval}, T_{risks}$)
8: **end if**
9: $T_{consent} \leftarrow \text{SignWithTwinKey}(A, T_{eval}, \text{timestamp})$
10:
11: **Phase 2: Human Verification**
12: notification $\leftarrow$ PrepareNotification($A, T_{eval}, T_{confidence}$)
13: SendToUser($U$, notification, timeout $= 24h$)
14: $U_{response} \leftarrow \text{WaitForResponse}(U, \text{timeout})$
15: **if** $U_{response} = \text{NULL}$ **then**
16: **Return** TIMEOUT, DefaultAction($A$)
17: **end if**
18:
19: **Phase 3: Consensus Formation**
20: **if** $U_{response}.\text{decision} = \text{APPROVE}$ **then**
21: $U_{consent} \leftarrow \text{SignWithUserKey}(A, U_{response}, \text{timestamp})$
22: consensus $\leftarrow$ CombineConsents($T_{consent}, U_{consent}$)
23: block_hash $\leftarrow$ BlockchainRecord(consensus)
24: ExecuteAction($A$, consensus, block_hash)
25: **Return** SUCCESS, block_hash
26: **else**
27: override_reason $\leftarrow U_{response}.\text{reason}$
28: LogOverride($U, T, A$, override_reason)
29: $T.\text{UpdatePreferences}(\text{override\_reason})$ ▷ Learn from override
30: **Return** USER_OVERRIDE, override_reason
31: **end if**

---

## B.6 Data Retention and Right to Erasure

Our data retention policies implement cryptographically-enforced expiration where data automatically becomes inaccessible after predetermined periods unless explicitly renewed through user consent. This system employs forward-secure encryption where encryption keys are automatically deleted after expiration, ensuring that even archived data cannot be recovered. The retention framework operates at multiple granularities, with conversation snippets expiring after 30 days, preference patterns after 90 days, and core personality models after one year unless actively renewed.

The right to erasure goes beyond simple deletion to implement cryptographic shredding that renders all derived data permanently inaccessible. When users invoke their right to be forgotten, the system not only deletes their primary data but also retrains affected models to remove their statistical influence. This process employs machine unlearning techniques that provably eliminate the user's contribution to aggregate models while maintaining performance for remaining users. Furthermore, our system maintains deletion certificates that provide cryptographic proof of erasure, giving users verifiable confirmation that their data has been permanently removed.

---

**Algorithm 11** Cryptographic Data Shredding with Verified Unlearning

---

1: **Input:** User $U$, Deletion request $D$, Verification key $V$
2: **Step 1: Immediate Access Revocation**
3: RevokeAllTokens($U$)
4: DisableDigitalTwin($U$.twin_id)
5: NotifyConnectedUsers($U$.connections)
6:
7: **Step 2: Cryptographic Key Destruction**
8: $keys \leftarrow$ GetAllUserKeys($U$)
9: **for** each $k \in keys$ **do**
10:     OverwriteMemory($k$, random, passes $= 7$)
11:     DestroyHSMEntry($k$.hsm_id)
12: **end for**
13:
14: **Step 3: Machine Unlearning**
15: affected_models $\leftarrow$ IdentifyAffectedModels($U$)
16: **for** each model $M$ in affected_models **do**
17:     influence $\leftarrow$ ComputeInfluence($U, M$)
18:     **if** influence $> \tau$ **then**
19:     $M' \leftarrow$ RetrainWithout($M, U$.training_data)
20:     **else**
21:     $M' \leftarrow M -$ influence $\cdot \nabla_U M$         $\triangleright$ Gradient-based unlearning
22:     **end if**
23:     VerifyUnlearning($M', U$)         $\triangleright$ Statistical test
24:     DeployUpdatedModel($M'$)
25: **end for**
26:
27: **Step 4: Audit and Certification**
28: audit_log $\leftarrow$ GenerateAuditTrail($U, D$)
29: deletion_proof $\leftarrow$ ZKProof(DataAbsence($U$))
30: certificate $\leftarrow$ Sign(audit_log, deletion_proof, $V$)
31: BlockchainRecord(certificate)
32: **Return** certificate, blockchain_hash

---

### B.7 PRIVACY METRICS AND CONTINUOUS MONITORING

Our privacy framework implements comprehensive metrics that continuously monitor privacy protection effectiveness across multiple dimensions. Re-identification risk is assessed through regular adversarial testing where red teams attempt to link anonymized profiles back to individuals using various attack strategies. These tests employ state-of-the-art linkage attacks, membership inference attempts, and attribute inference techniques to ensure our defenses remain robust against evolving threats. The system maintains a real-time privacy dashboard that tracks key metrics including k-anonymity levels, differential privacy budget consumption, and information leakage rates across all components.

Behavioral uniqueness metrics quantify how distinguishable individual users remain after our anonymization processes. We employ information-theoretic measures to assess the entropy of user behavior patterns and ensure sufficient uncertainty to prevent tracking. The monitoring system automatically triggers additional privacy protections when uniqueness scores exceed thresholds, dynamically adjusting noise levels and generalization parameters to maintain target privacy levels. This adaptive approach ensures consistent privacy protection even as user behavior patterns evolve and new interaction modalities are introduced.

The watermark detection system undergoes continuous calibration through automated testing against adversarial paraphrasing attacks. We maintain a library of evasion techniques including synonym substitution, sentence restructuring, back-translation, and GPT-based paraphrasing, constantly evaluating our watermark robustness against these attacks. Detection thresholds are dynamically adjusted to maintain false positive rates below $10^{-6}$ while maximizing true positive rates, with regular updates to handle new attack vectors as they emerge.

### B.8 REGULATORY COMPLIANCE AND LEGAL FRAMEWORK

Our privacy framework ensures comprehensive compliance with international data protection regulations including GDPR, CCPA, and emerging AI governance frameworks. For GDPR compliance, we implement privacy by design principles throughout the system architecture, ensuring that privacy protection is not an afterthought but a fundamental design consideration. Our lawful basis for processing relies on explicit consent with granular opt-in mechanisms for different data uses. The system maintains detailed records of processing activities, implements data protection impact assessments for new features, and ensures cross-border data transfers comply with adequacy decisions and standard contractual clauses.

For employment contexts, our framework adheres to EEOC guidelines and international fair hiring practices through algorithmic auditing and bias mitigation. The system implements disparate impact testing that continuously monitors hiring recommendations for discriminatory patterns across protected categories. When bias is detected, the system automatically adjusts cognitive module weights and decision thresholds to restore fairness while maintaining prediction accuracy. All employment-related decisions generate detailed explanations that can be reviewed by human resources professionals and challenged by candidates through established appeal processes.

The legal framework extends to intellectual property protection for generated content, implementing clear ownership models and usage rights for twin-generated text. Our terms of service establish that users retain ownership of their digital twin's outputs while granting necessary licenses for system operation. The watermarking system provides forensic capabilities for intellectual property disputes, enabling verification of content origin and protecting against false attribution claims. We maintain comprehensive insurance coverage for privacy breaches and implement incident response procedures that ensure rapid notification and remediation in case of any privacy violations.

### B.9 FUTURE DIRECTIONS AND RESEARCH CHALLENGES

While our current privacy framework provides strong protection against known threats, several research challenges remain for future development. Adversarial machine learning attacks that attempt to extract training data or manipulate model behavior require ongoing defensive research and regular model hardening. We are investigating certified defenses that provide provable robustness guarantees against specific attack classes. The emergence of quantum computing threatens current cryptographic protections, necessitating migration to quantum-resistant algorithms for long-term data protection. Our roadmap includes systematic replacement of classical cryptographic primitives with lattice-based and hash-based alternatives that resist quantum attacks.

The intersection of privacy and explainability presents fundamental tensions that require novel technical solutions. Users deserve explanations for their digital twins' decisions, but detailed explanations

can leak private information about the training process and other users. We are developing privacy-preserving explanation techniques that provide meaningful insights while maintaining differential privacy guarantees. This includes research into synthetic explanations that capture decision logic without revealing specific training examples and federated explanation generation that aggregates insights across multiple users while preserving individual privacy.

The scalability of privacy-preserving technologies remains a significant challenge as user bases grow and interaction complexity increases. Homomorphic encryption and secure multi-party computation introduce substantial computational overhead that can impact system responsiveness. Our research focuses on optimizing cryptographic protocols for specific computation patterns in digital twin interactions and developing hardware acceleration for privacy-preserving operations. We are also investigating selective privacy relaxation where users can choose to trade privacy for performance in non-sensitive contexts while maintaining strong protection for critical interactions.

### B.10    CONCLUSION

The comprehensive privacy protection framework presented here establishes a new standard for protecting user privacy in psychologically-authentic digital twin systems. Through the integration of differential privacy, cryptographic watermarking, federated learning, and sophisticated consent mechanisms, our framework ensures that CogniPair can deliver meaningful social pairing while respecting user privacy and autonomy. The multi-layered approach provides defense in depth against various attack vectors while maintaining the utility necessary for practical deployment. As digital twins become increasingly sophisticated and prevalent, the privacy protections outlined here will be essential for maintaining user trust and preventing potential harms. Our ongoing research continues to strengthen these protections while exploring new frontiers in privacy-preserving artificial intelligence, ensuring that the benefits of digital twin technology can be realized without compromising fundamental privacy rights.

