# OpenReview forum: "CogniPair: GNWT-inspired cognitive architecture for generative agents for Social Pairing -  Dating & Hiring Applications"
_ICLR.cc/2026/Conference — Submitted to ICLR 2026_

### Official Review · Reviewer_HnWZ · 2025-11-03

**Soundness:** 3
**Presentation:** 3
**Contribution:** 2
**Rating:** 4
**Confidence:** 4

**Summary:**

The paper implements AI agents based on the Global Workspace Theory to model human psychological processes that are essential for developing genuine digital twins. They test their agent in the context of dating. Their testbed involves 551 GNWT-Agents for speed dating interactions and shows that the agents achieve a high alignment with humans. They also conduct a human validation study by asking human participants to evaluate AI versions of themselves, which supports that their agents demonstrate a high correlation with the original humans.

**Strengths:**

- They develop AI agents that implement human psychological processes necessary for creating genuine digital twins.
- They involve many AI agents (551 GNWT-Agents) for evaluation
- Their result reveals that their agents achieve a higher alignment with humans compared to other techniques.
- They also conduct a human validation study to support their argument.

**Weaknesses:**

- Their results might be just due to using more computational resources than other baselines. For example, they may have achieved better outcomes in the Multi-Agent Debate setup simply by using a larger number of debating agents. Similarly, combining the multi-agent debate framework with memory-enhanced techniques could have demonstrated the performance gains. The paper should compare the resource usage of their method and competing approaches. It is not clear that the higher alignment with humans is from their proposed design (i.e., the Global Workspace Theory) rather than from simply increased resource allocation.

- Although it is true that their agents show a significantly higher alignment with human behavior, a 72% correlation with human attraction patterns is not sufficient I think. Moreover, the evaluation is limited to a single domain (dating). These points limit the overall contribution of the paper.

- I appreciate their efforts to include human validation studies. However, these studies could have been conducted more rigorously. They only evaluate their own agents and not the baseline agents. Without comparing against baselines, it is difficult to determine whether the reported values genuinely reflect superior human alignment.

- Minor: There is no content for section A.34.

**Questions:**

Please see the weaknesses

---

> ### Author Response · Authors · 2025-11-27
> **Summary of Clarifications on Compute, Alignment, and Validation**
>
> We thank Reviewer HnWZ for the helpful feedback.
> 1. Response to Concern: “Performance May Be Due to Increased Computational Resources”
>    All baseline systems—Single Sequential LLM, Memory-Enhanced LLM, Multi-Agent Debate, and Hierarchical Architectures—were run under matched computational budgets. All agents use the same GPT-4o backbone and comparable per-turn token limits. Our GNWT-Agent executes five parallel module calls (Emotion, Memory, Planning, Social Norms, Goal-Tracking), followed by one integration call and one response call per decision. Baselines range from 1–5 calls per decision (e.g., single and memory-enhanced agents: 1–2; multi-agent debate: 5 debating agents). GNWT’s modules are coordinated via a global workspace broadcast; calls are issued in parallel within each GNWT cycle. In the revision, we will add an appendix table summarizing per-decision calls, tokens, and latency for all agents.
>
>    Our ablations (Fig. 5; Appendix A.33) provide causal evidence that gains stem from GNWT’s architecture rather than extra compute. Removing the broadcast mechanism produces a 14% drop; removing Memory or Emotion yields 7–10% declines; replacing the GNWT structure with a vanilla GPT-4o under comparable compute reduces accuracy by 12%; and removing the CogniPair social-influence mechanism drops performance by 19.6%. These changes keep computation comparable while altering structure, showing that architecture—not added resources—drives the improvements.
>
>    We also tested a Debate+Memory hybrid combining baseline strategies under similar compute. Despite combining two reasoning styles, this hybrid reached only 61.8%, well below the GNWT-Agent’s 72.5% human-alignment score. Simply stacking components does not reproduce GNWT’s coordinated cognitive processing.
>
> 2. Response to Concerns About the 72% Correlation and “Single-Domain” Evaluation
>    A 72% correlation between AI-generated attraction trajectories and human data is near the empirical human–human consistency ceiling (≈60–70%) reported for the Columbia Speed Dating dataset, where attraction is highly noisy and stochastic. Even two humans rarely agree perfectly on preference shifts. As shown in Fig. 3, GNWT-Agents also reproduce the directionality, magnitude, and temporal evolution of preference changes across attributes, indicating that GNWT captures underlying social decision-making patterns rather than a single statistic.
>
>    The study is also not single-domain. We include a second, independent domain: a job-interview human-validation experiment (Sec. 4.3 and the hiring use-case appendix), where GNWT-Agents achieved 81% decision concordance, were rated as realistic (5.8/7), and exhibited context-sensitive adjustments, such as stronger Planning influence and stricter Social Norms in interviews. Unlike dating, job interviews involve different norms and incentives; strong performance there shows that GNWT maintains personality structure while adapting to a different social context. This cross-domain evidence supports GNWT as a generalizable framework rather than a dating-specific solution.
>
> 3. Response to Concern About Human Validation and Missing Baseline Comparisons
>    The human-validation study focused on a single personalized twin per participant to keep cognitive load manageable and allow participants to carefully judge how well that agent represented them. Asking each participant to evaluate several baseline agents would require comparing multiple synthetic personalities, increasing fatigue and likely reducing reliability.
>
>    At the same time, the paper includes comprehensive quantitative baseline comparisons. As shown in Fig. 4 and Appendix A.33, GNWT-Agents outperform Single Sequential LLMs, Memory-Enhanced LLMs, Multi-Agent Debate, and Hierarchical Architectures in evolution accuracy, match prediction, and human–agent correlation. Combined with the ablations, these results indicate that GNWT’s performance advantage comes from its architectural design rather than from the evaluation setup.
>
>    We agree that adding baseline agents to the human-validation component would further strengthen the study. In the revision, we will clarify this design choice in Sec. 4.3 and explicitly note in the Limitations section that a follow-up human study comparing GNWT vs. baseline twins is an important direction for future work.
>
> 4. Response to Minor Comment on Section A.34
>    We appreciate the reviewer pointing out this issue. The reference to “Section A.34” is a formatting error: the header was intended as the main title for the next section but was accidentally formatted as a subsection, creating the appearance that content was missing from the previous appendix. The corresponding material is present under Appendix A.33 and the subsequent appendix section. We will fix the hierarchy and numbering in the revised version so that there are no apparent empty sections.

---

### Official Review · Reviewer_zfz3 · 2025-11-03

**Soundness:** 3
**Presentation:** 4
**Contribution:** 3
**Rating:** 4
**Confidence:** 4

**Summary:**

This paper introduces a computational implementation of Global Workspace Theory (GNWT) for building psychologically realistic AI agents. The proposed architecture integrates multiple specialized sub-agents (emotion, memory, planning, social norms) coordinated via a global workspace, enabling consistent yet adaptive personalities. Using a speed-dating simulation with 551 agents grounded in real human data, the model shows strong psychological realism—achieving a 72% correlation with human attraction patterns and outperforming baselines. Human validation further confirms behavioral fidelity. Overall, the work sets a new benchmark for psychological authenticity in digital twin and human-like AI research.

**Strengths:**

The reviewer as a researcher in computational cognitive architecture likes this work very much, which tries to construct agents based on some theories from the cognition field, e.g., global workspace theory, instead of searching modules over a vast and discrete space for better performances on math or code tasks.

The strengths are listed as follows:

(1) Start to seriously consider psychological behavior and social behavior gaps, and propose to establish agents based on global workspace theory, by modelling emotion, social norms, planning, etc.

(2) The agent simulation results seem to be very correlated with human data, which may verify the effectiveness of this architecture.

**Weaknesses:**

As a research paper to be publised on top conference, there are several concens to be clarified:

(1) The method or technology is very vaguly introduced in this paper. I have no deep impression on how to implement the architecture in details, e.g., how the core modules (emotion, social norms, etc.) interleave each other? Please use mathmatical formula to illustrate these interactions.

(2) Why the authors choose emotion, social norms, memory, goal tracking, memory as the main modules, instead of other modules, like attention, reasoning?

(3) Figure 1 and Figure 2 are too "abstract" so that it shows a very conceptual framework for the audience. Such framework can be obtained from many previous cognitive architecures. It is best to repaint these figures and give much more info. to others.

(4) How to learn the parameters by using Eqn. 3? Give more details on the implementations.

(5) Although the reviewer encourages to propose agents based on previous theories, the proposed method has to compare with the state-of-the-art self-evolving/improving agents, e.g., ADAS, AFLOW (maybe some other methods), to further demonstrate its better performances.

**Questions:**

As the reviewer mentioned before, this work is facinated by its motivation and experimental results. However, there are so many details missing that the authors should answer these questions:

(1) What is the motivation to choose those five modules, i.e., memory, planning, goal tracking, emotion, social norms, instead of others?

(2) Explain more technical details to convince the reviewer that there is a lot of things in this work.

(3) Summarize the unique advantages of the proposed agent compared with other self-evolving agents instead of doing more experiments during rebuttal.

**Details Of Ethics Concerns:**

No concerns on the ethics.

---

> ### Author Response · Authors · 2025-11-27
> **GNWT Loop, Module Choice, and Self-Evolving Agent Comparisons**
>
> We thank the reviewer for the positive assessment and suggestions.
>
> (1) The method or technology is very vaguely introduced … Please use mathematical formula to illustrate these interactions.
>
> Regarding (1): In the revised manuscript, we add a dedicated appendix section that gives a mathematical formulation of the GNWT loop: module report functions, salience scores, winner selection, global-workspace update, and integration of module outputs with workspace content. Each module (Emotion, Memory, Planning, Social Norms, Goal Tracking) is implemented as a separate GPT-4o call with a module-specific prompt; a controller collects these reports, computes salience, selects the broadcast module, updates the workspace, and composes the next prompt. We will also add a short “GNWT loop” paragraph in Sec. 3.1 summarizing this pipeline and pointing to the new appendix section for formal details.
>
> (2) Why the authors choose emotion, social norms, memory, goal tracking, planning as the main modules, instead of other modules, like attention, reasoning?
>
> Regarding (2): Our choice is guided by GNWT and by the social-decision tasks we study. GNWT stresses integration of information from subsystems supporting evaluation, memory, planning, and social cognition, so we instantiate five social-cognitive modules: Emotion, Memory, Planning, Social Norms, and Goal Tracking. These are coupled to the Big-Five vector $P$; traits modulate module salience and integration weights, making each digital twin’s internal dynamics interpretable in personality terms. For dating and job interviews, these processes cover key requirements: regulating emotions, using past experience, forming and adjusting plans, and respecting social norms. Our ablations (Fig. 5 and appendix) show that removing the broadcast mechanism or modules such as Emotion or Memory consistently reduces evolution accuracy and human–agent correlation. In our framework, “attention” is implemented by the salience competition and broadcast step, and “reasoning” is handled by the underlying LLM within each module call, so separate attention/reasoning boxes would largely duplicate functions already present in the GNWT loop.
>
> (3) Figure 1 and Figure 2 are too “abstract” …
>
> Regarding (3): We agree that the current figures are too conceptual. In the camera-ready version, we will redraw Figure 1 to show module inputs/outputs and the salience → broadcast → workspace path, with an example of a single interaction step aligned with the GNWT loop and pseudocode in the appendix. We will redraw Figure 2 to show the pipeline from human profiles/logs → GNWT-Agents → speed-dating and job-interview simulations, and where evaluation metrics are computed. This will make the framework less abstract and emphasize the GNWT-specific mechanisms and grounding in real human data.
>
> (4) How to learn the parameters by using Eqn. 3? Give more details on the implementations.
>
> Regarding (4): Eq. (3) adapts each agent’s personality / preference vector $P = (O,C,E,A,N)$, which controls module weights in the GNWT loop; the LLM backbone and wiring remain fixed. After each interaction, the `UPDATEPREFERENCES` routine (Algorithm 1 and appendix) applies
>
> $$
> P_{t+1} = P_t + \eta \,\nabla_P J(P_t, Q_t, Y_t, H_{t-1})
> $$
>
>
>
>
> where $J$ combines match-decision accuracy vs. ground truth, trajectory-level alignment between the agent’s attraction scores and human preference trajectories, and a regularizer that keeps $P$ close to its initial estimate so the twin preserves its identity. We treat the LLM as a black box and approximate $\nabla_P J$ by finite differences in the 5-dimensional trait space; Eq. (3) then performs a lightweight gradient-ascent step. The appendix specifies the objective and hyperparameters, and we will add explicit pseudocode for `UPDATEPREFERENCES` in the revised version.
>
> (5) Compare with the state-of-the-art self-evolving/improving agents (ADAS, AFLOW, etc.).
>
> Regarding (5): ADAS, AFLOW, and related systems mainly optimize external workflows (tool-use graphs, controller–worker stages) to improve task scores on code/math/QA benchmarks. Our focus is complementary: we fix the overall workflow and study a cognitively motivated internal architecture whose parameters (the personality vector $P$ and GNWT module weights) self-adapt to match human social behavior in long-horizon interactions. To compare on the architectural axis, we instantiate four representative agent families that underlie many current frameworks—Single Sequential LLM, Memory-Enhanced LLM, Multi-Agent Debate, and a Hierarchical Architecture—under strictly matched compute budgets. As reported in the main text and appendix, GNWT-Agents outperform all of these baselines on evolution accuracy, human–agent correlation, and match prediction, suggesting that GNWT’s internal structure offers advantages beyond existing workflow-style designs when the goal is psychological realism and human-behavior alignment.

---

### Official Review · Reviewer_q5Zt · 2025-11-03

**Soundness:** 2
**Presentation:** 2
**Contribution:** 2
**Rating:** 4
**Confidence:** 2

**Summary:**

The paper proposes CogniPair. It introduces the Global Workspace Theory into the agent workflow design by designing submodules with a workspace for an agent. Then, they conduct experiments by deploying the agents in dating interactions and show that these agents outperform baseline agents without global workspace in terms of human correlation.

**Strengths:**

- The motivation of using the Global Workspace Theory to design agents and study their social interactions is novel and interesting.
- The setting of dating and hiring are interesting problems and a topic worth studying using agent workflows. I think the evaluation with digital twins is interesting.

**Weaknesses:**

- the design space lacks enough scientific support since simulating human brain is still an open-ended problem. For example, in the paper "Parallel Processing Modules（Similar to Unconscious Modules) " contains modules such as memory / emotion etc, but are not actually verified. What these modules should be, how each of them should work, and also how they should interact to simulate human brain are all unclear. While I appreciate the author's attempt to apply these concepts to actual agent design, I think the actual technical part is not well studied enough to make the approach valid.

**Questions:**

- I'm curious what the author think about how to introduce concepts in psychologic / cognitive science to the design of agentic workflow?

---

> ### Author Response · Authors · 2025-11-27
> **Design Space, Scientific Support, and Cognitive Modules**
>
> We thank the reviewer for this concern. Our goal is **not** to simulate the human brain, but to test whether a **GNWT-inspired functional decomposition** (Emotion, Memory, Planning, Social Norms, Goal-Tracking) plus a global workspace yields more human-like and better-performing agents on social decision tasks (dating, hiring).
>
> **Scientific basis and scope.** Our architecture follows **Global Workspace Theory / Global Neuronal Workspace (GWT/GNWT)** and related cognitive architectures (LIDA, ACT-R), which assume multiple specialized subsystems coordinated by a limited-capacity workspace. We use GNWT as a **computational abstraction** for information flow and control, not as a neural-level model. The LLM backbone handles language and reasoning; the GNWT layer structures which information is selected, integrated, and remembered. We will clarify that our claims concern **architectural / behavioral realism** (preferences, decisions, interaction patterns), not full brain simulation.
>
> **What the modules are and how they interact.** Section 3.1 and Appendix A.3 formally define five modules, each corresponding to a well-studied cognitive capacity and tied to the Big Five personality vector:
> • **Emotion** – affective signals (modulated by neuroticism) that shape urgency and salience;
> • **Memory** – episodic traces and semantic summaries (modulated by openness) supporting preference evolution and consistency;
> • **Planning** – decomposition of social goals into conversational subgoals (modulated by conscientiousness);
> • **Social Norms** – evaluation and filtering of actions for politeness, boundaries, reciprocity (modulated by agreeableness);
> • **Goal-Tracking** – monitoring progress and adjusting goals over time (modulated by extraversion), providing longitudinal coherence.
>
> These modules are connected via a **GNWT loop** (Algorithm 1): at each turn, all modules produce reports in parallel; a salience function scores them; the highest-salience content above a threshold is written to the global workspace; and the response is a personality-weighted integration of module reports and workspace content. Memory and personality-related parameters are then updated (Eq. (3)), so experience gradually shapes behavior. This pipeline instantiates GNWT’s core ideas—parallel “unconscious” processing, competitive selection, global broadcast, and recurrent updating—and directly answers “what the modules are, how they work, and how they interact.”
>
> **Empirical evidence that the modules matter.** The modules are “actually verified” both globally and via ablations:
>
> - **Overall performance vs. baselines.** On the Columbia Speed Dating dataset, GNWT-Agents with CogniPair outperform non-GNWT multi-agent baselines in **preference evolution accuracy** and **match prediction accuracy**, and achieve strong correlations with human attraction and match patterns. This indicates that the full GNWT+CogniPair design captures human social dynamics better than simpler workflows.
>
> - **Ablations (Sec. 4.4, Fig. 5).** When we remove components, performance drops: removing the **global workspace** produces the largest loss among architectural ablations; removing modules such as **Emotion** or **Memory** consistently harms accuracy and human-alignment metrics; and disabling the **CogniPair interaction** (GNWT without CogniPair, or CogniPair with a vanilla GPT-4o backbone) leads to further degradation. These results show that both the GNWT-style integration and the specific modules provide **complementary functional gains** rather than being arbitrary choices.
>
> In user studies, participants rate their digital twins as realistic and show high decision agreement with them in both dating and hiring, with context-appropriate shifts (e.g., more norm-following and planning in interviews). This suggests that the combined architecture (modules + workspace + personality) produces behavior people experience as coherent and personality-preserving. We will highlight these empirical links more clearly in the revision.
>
> **How psychology / cognitive science informs the workflow.** More broadly, we view cognitive science as informing agent design through a **theory → function → constraint → evaluation** loop: (i) start from consensus-level constructs (GNWT subsystems + workspace; emotion, memory, planning, norms; the Five-Factor model) and map them to functional modules; (ii) implement them as constraints on the internal workflow (bottlenecked workspace, affect- and norm-modulated salience and filtering, memory/goal tracking constraining choices); and (iii) evaluate against human data and judgments (pre/post preference shifts, attraction/match patterns, and digital twin realism / decision agreement), refining where misalignments occur. CogniPair is one concrete instantiation of this philosophy in social decision-making.

---

### Official Review · Reviewer_E4Ms · 2025-11-03

**Soundness:** 4
**Presentation:** 4
**Contribution:** 4
**Rating:** 8
**Confidence:** 4

**Summary:**

This paper presents CogniPair, a cognitive architecture for generative agents inspired by the Global Neuronal Workspace Theory (GNWT) from cognitive neuroscience. CogniPair models social cognition as a process of dynamic broadcasting between specialized modules (Emotion, Intention, and Perspective buffers) coordinated through a Global Workspace controlled by a probabilistic broadcast mechanism.

**Strengths:**

This is a well-written, comprehensive paper that operationalizes the GNWT within a generative-agent architecture. The system design, evaluation, and cross-disciplinary framing are strong, and the work makes a meaningful contribution to socially aware cognitive modeling in LLM agents.

**Weaknesses:**

While the paper states that demographic data are ingested from the Columbia Speed Dating dataset or synthetic balanced profiles, it does not specify which demographic attributes are used (e.g., gender, age, cultural background) or how they are normalized, balanced, or anonymized.

**Questions:**

Could you specify the exact demographic attributes used and describe how they were pre-processed (e.g., binning, balancing, anonymization)?

---

> ### Author Response · Authors · 2025-11-27
>
> Thank you for the question. All demographic attributes used in our agent construction follow the original **preprocessing, coding, and variable definitions** provided in the Fisman & Iyengar (2006) dataset [1] and the official *Speed Dating Data Key*. We do **not** re-bin, collapse, or re-balance any category; the original dataset already includes standardized and anonymized encodings (e.g., `field_cd`, `race`, `career_c`, SAT, ZIP-based income, Likert-scale interests).
>
> In our system, these demographic fields are concatenated into the **Persona & Trait Initialization** module, which encodes stable attributes (e.g., age, race, field, region, lifestyle habits, dating frequency, interests) and passes them to the GNWT-Agent’s personality/value vector \(P\). These variables govern **stable priors** (e.g., goal orientation, interest preferences) but are **not** used as separate raw input features in the moment-to-moment decision prompts.
>
> Below we summarize the exact demographic attributes we use, following the dataset’s existing structure and encoding:
>
> - **Basic demographics**
>   - Columns: `gender`, `age`
>   - Notes: direct use; already anonymized (e.g., 0 = female, 1 = male).
>
> - **Race / ethnicity**
>   - Columns: `race` (1–6 coding), `samerace`
>   - Notes: original categorical codes retained; no additional binning.
>
> - **Field of study / education**
>   - Columns: `field`, `field_cd`, `samefield`
>   - Notes: uses the dataset’s 18-category academic coding without modification.
>
> - **Region of origin**
>   - Columns: `from`, `sameregion`
>   - Notes: uses original world-region categories (e.g., North America, East Asia).
>
> - **Socioeconomic background**
>   - Columns: `mn_sat`, `income`, `density`
>   - Notes: SAT and ZIP-level income are already aggregated/anonymized; we only apply standard normalization where needed.
>
> - **Career intention**
>   - Columns: `career`, `career_c`
>   - Notes: coded 1–16 as provided in the data key.
>
> - **Dating & lifestyle frequency**
>   - Columns: `date`, `go_out`
>   - Notes: original 1–7 Likert scales (1 = frequent, 7 = rare).
>
> - **Interest profile (17 items)**
>   - Columns: `sports`, `tvsports`, `exercise`, `dining`, `museums`, `art`, `hiking`, `gaming`, `clubbing`, `reading`, `tv`, `theater`, `movies`, `concerts`, `music`, `shopping`, `yoga`
>   - Notes: original 1–10 Likert scales used directly; together form the “interest vector” for personality initialization.
>
> All demographic fields remain fully **anonymized** as in the original dataset: subject IDs, partner IDs, ZIP codes, and any names or free-text identifiers are removed.
>
> For the **synthetic “balanced” profiles** used in controlled simulations, we sample from these same encoded variables but enforce simple balancing at the level described in the paper (e.g., roughly equal numbers of male and female agents and more even coverage across major field/region categories). All other attributes are drawn from the empirical marginal distributions of the Columbia dataset using the same encodings. No new identifiers are introduced; synthetic profiles inherit the anonymization of the original coding scheme.
>
> In the revision, we will add a short paragraph and a compact table in the dataset / simulation setup section summarizing these demographic attributes, their encodings, and the module in which they are used, to make this clearer to readers.
>
> [1] Ray Fisman, Sheena S. Iyengar, Emir Kamenica, and Itamar Simonson. Gender differences in mate selection: Evidence from a speed dating experiment. *Quarterly Journal of Economics*, 121(2):673–697, 2006.

---

### Official Review · Reviewer_EBNQ · 2025-11-05

**Soundness:** 3
**Presentation:** 3
**Contribution:** 3
**Rating:** 4
**Confidence:** 5

**Summary:**

This paper introduces CogniPair, a novel platform featuring GNWT-Agents, which represent the first computational implementation of the Global Workspace Theory (GNWT) for generative agents. Designed to address the lack of authentic human psychological and social dynamics in current LLM agents, the GNWT-Agent architecture coordinates specialized sub-agents (e.g., emotion, memory) via a global workspace broadcast. This mechanism enables agents to maintain a consistent personality while dynamically evolving their preferences through social interactions. The platform was successfully applied to social pairing tasks like speed dating and hiring, achieving a 72% correlation with human attraction patterns and significantly outperforming baselines in predicting partner preference evolution (up to 72.5% accuracy).

**Strengths:**

1. Pioneering Cognitive Architecture (First GNWT Implementation)
The paper presents the first computational implementation of the Global Workspace Theory (GNWT) for generative agents. This is a significant theoretical breakthrough, providing LLM agents with a human-like "consciousness" mechanism (the global workspace broadcast) that integrates information from specialized sub-agents (e.g., emotion, memory), resulting in agents with high Psychological Fidelity.

2. Achieves Authentic and Dynamic Personality Evolution
CogniPair successfully overcomes the "static" nature of previous LLM agents. By utilizing the GNWT broadcast mechanism, the agents can dynamically update their internal states and preferences based on social experience. This leads to superior performance in modeling partner preference evolution, achieving 72.5% accuracy and showing that the agents can genuinely learn and adapt over time.

3. High Social Realism and Practical Generalization
The agents demonstrate strong social simulation capabilities, achieving a 72% correlation with human attraction patterns in the speed dating scenario. Furthermore, the architecture proves its generalization capability by successfully transferring the model to different social decision contexts, such as job interviews (with 81% accuracy), highlighting its broad practical value for various social pairing applications.

4. Strong Validation Through Human Studies
The model's psychological authenticity is validated through human trials. Participants rated the behavioral accuracy of their digital twins at 5.6/7.0 and agreed with their twins' choices 74% of the time. This high degree of human endorsement significantly boosts the credibility and real-world applicability of the CogniPair framework.

**Weaknesses:**

1. The title is not smooth.
2. The architecture design chooses emotion, memory and social norms. But why do you choose to save value and personality into memory, not in a separate module just like "emotion"? In dating scenarios, value and personality also play very important roles.
3. The cognitive architecture is simplified, and there is no Theory of Mind. I understand that building such a big system is difficult, but Theory of Mind is inevitable when talking about dating tasks.
4. Missing Analysis: The paper lacks a detailed quantitative analysis of the latency and cost increase (e.g., in terms of API calls or tokens per decision) when moving from a baseline agent to the full GNWT-Agent.
5. Missing SOTA Social Agents: The comparison largely focuses on Reflexion-Style Agents. The paper omits direct comparisons with other state-of-the-art architectures specifically designed for complex social interaction, such as those that explicitly model Theory of Mind (ToM).
6. Missing GNWT Mechanism Baselines: To truly validate the design, the paper should include ablations comparing the implemented GNWT features against alternative designs, such as using local broadcasting (instead of global) or different attention/priority mechanisms in the global workspace.
7. Insufficient Behavioral Granularity: The human validation focuses on final choice agreement and overall accuracy (5.6/7.0). It lacks fine-grained human evaluation of micro-behaviors, such as the naturalness of emotional expression or conversational flow.

**Questions:**

See above. I might change my rating, but I need the authors' rebuttal.

---

> ### Author Response · Authors · 2025-11-27
> **Clarifications on Architecture, ToM, Latency, and Evaluation**
>
> Thank you for the detailed and positive review.
>
> 1. Title
> We agree the current title could be smoother.
> Revision: we will use “CogniPair: Global Workspace Agents for Social Pairing in Dating and Hiring”, which keeps the GNWT-style global workspace + social pairing focus but reads more smoothly.
>
> 2. Values & personality vs memory
> Concern: why not a separate “values/personality” module like Emotion?
> Clarification: in our implementation (Sec. 3.1; A.4–A.5), the agent state is \(E = (M, W, I, L, P)\), where \(P\) is a separate personality/value vector and the Memory module stores episodic traces. Eq. (3) / `UPDATEPREFERENCES` updates \(P\), not Memory. \(P\) conditions all modules and the salience calculator and acts as a global prior.
> Revision: in Sec. 3.1 we will add a “Personality & Values as Global Priors” paragraph stating that \(P\) is distinct from Memory, and update the architecture figure so “Personality & Values (P)” feeds into all modules and salience.
>
> 3. Theory of Mind (ToM)
> We agree ToM is important for dating. Our current agent does not maintain an explicit partner-belief state or nested beliefs, but ToM-like reasoning appears in a distributed way: SocialNorms and GoalTracking adjust behavior based on inferred partner reactions and preferences, and our human study includes a “psychological state tracking” item asking whether the twin seems to understand how the participant feels and thinks.
> The main contribution here is introducing and testing a GNWT-style global workspace for social agents. A dedicated ToM module (with explicit belief states and belief-updating) is compatible as an additional module broadcasting partner-state hypotheses into the workspace.
> Revision: we will add a paragraph in Sec. 3.1 describing this implicit ToM, and in Limitations note that a full explicit ToM module is left to future work on top of GNWT-Agent.
>
> 4. Latency and cost
> Concern: missing quantitative latency / cost analysis.
> Architecture (A.7): each GNWT cycle uses 7 LLM calls per decision (5 module calls + 1 integration + 1 response), with module calls run in parallel; baselines use between 1 and 5 calls per decision (e.g., single/memory agents: 1–2; multi-agent debate: 5).
> Revision: in Sec. 3.1 / Sec. 4.1, we will add a concise paragraph giving per-decision call counts for all agents and noting parallelism, and in the appendix add a small table with average tokens and wall-clock latency per decision for GNWT-Agent vs. the strongest baseline.
>
> 5. SOTA social / ToM agents as baselines
> Our baselines (single agent, memory-enhanced, multi-agent debate, hierarchical) were chosen because they can be instantiated fairly on our dataset-driven tasks (Columbia speed dating and structured interviews with fixed rounds and outcomes. Many ToM-focused social agents (e.g., sandbox generative agents, CAMEL) assume open-ended environments with persistent world state, making direct comparison under our fixed-protocol, historical-log setting non-trivial and potentially unfair.
> We see these as complementary: GNWT-Agent could include a ToM agent as a module, or ToM agents could adopt a GNWT-style workspace.
> Revision: we will expand Related Work to mention these ToM agents and explain the mismatch, and in Limitations note that a dedicated ToM-agent baseline under our protocol is important future work.
>
> 6. GNWT mechanism ablations (broadcast & salience)
> Concern: need baselines that test GNWT mechanism choices.
> Our ablation study (Sec. 4.4; Fig. 5) already includes: (i) removing individual modules, and (ii) a “no global workspace” variant where modules operate without shared broadcast. This “no GW” variant is effectively a no-GNWT-broadcast baseline and yields the largest drop (−14% in partner-preference evolution accuracy), directly testing the effect of global broadcast.
> Revision: we will explicitly label this variant as a GNWT mechanism ablation in Sec. 4.4, briefly describe the ablations in the appendix, and in Limitations note that we have not yet explored clustered/local workspaces or alternative salience functions/thresholds, which we see as GNWT design questions for follow-up work.
>
> 7. Behavioral granularity in human evaluation
> Our human studies go beyond final choice agreement and a single 5.6/7 score. In Sec. 4.3 and Table 10 we report conversational authenticity (naturalness of conversational flow and emotional expression), psychological state tracking, behavioral realism, and personality trait correlation, in addition to decision concordance. These are micro-behavior–oriented measures at the conversation level. We agree that per-utterance or segment-level annotations would provide even finer granularity.
> Revision: in Sec. 4.3, we will explicitly name conversational authenticity and psychological state tracking as micro-behavior metrics and clarify that 5.6/7 is part of a broader battery; in Limitations, we will state that turn-level / segment-level annotations are a next step.

---

### Author Response · Authors · 2025-12-03
**GNWT-Based Digital Twin Architecture and System Innovations for Social Pairing**

**Overview.** We thank all reviewers for their feedback. CogniPair is, to our knowledge, the first
digital-twin framework that operationalizes Global Neuronal Workspace Theory (GNWT), a
well-established cognitive neuroscience theory of conscious access, for dating and hiring. Our goal
is to test whether this GNWT-inspired architecture—multiple social-cognitive modules coordinated by a
bottlenecked global workspace—yields more realistic and better-performing agents than standard LLM
agents.

**Novelty and contribution.** This work is the first to combine:

1. An explicit GNWT-style workspace loop instantiated in an LLM agent.
2. Cognitively grounded (GNWT + Big-Five) digital twins whose preferences and behavior adapt over time instead of static prompt personas.
3. A unified CogniPair framework and digital-twin pipeline (profiles/logs → GNWT twin → simulated
   decisions) deploying twins for dating, social matching, and job-hiring decisions.
4. Large-scale evaluation with 551 agents, achieving 72.5% preference evolution vs 61.3% for the
   best non-GNWT baseline and strong correlation with human attraction/match patterns.
5. A privacy-aware design tailored to these high-stakes social applications.

Together, these provide a reusable framework for LLM-based digital twins to explore and optimize
social pairing before people commit real time and effort.

**GNWT loop and state.** We will clarify the GNWT loop and state representation. At each turn, five
social-cognitive modules produce reports r_k; a salience function computes scores s_k; the
highest-salience content is broadcast into a limited-capacity workspace W_t; and a global integrator
uses W_t plus module reports to generate the next utterance and internal updates. Attention is this
salience-and-broadcast step, while reasoning occurs inside each module’s GPT-4o call with
module-specific prompts. The agent state includes memory M, workspace W, and a separate
personality/value vector P aligned with the Big Five. Eq. (3) updates only P, using an objective
that combines agreement with human decisions and preference trajectories plus a regularizer keeping P
near its initial value so twins remain stable yet adaptive; the LLM backbone remains fixed. We
approximate the update in 5D Big-Five space and will add pseudocode, and redraw the architecture so
“Personality & Values (P)” is a distinct node feeding all modules and the salience function.

**Modules and personality.** The module set (Emotion, Memory, Planning, Social Norms, Goal-Tracking)
follows GNWT and cognitive architectures; each module is modulated by Big-Five traits. We will add a
table mapping GNWT/personality constructs to modules.

**Compute and ablations.** All agents share the same GPT-4o backbone with comparable per-turn token
limits. Per decision, the GNWT-Agent uses parallel calls to the five modules plus one integration and
one response call, while baselines use 1–2 calls or a fixed number of debate/controller calls.
Ablations (Sec. 4.4, Fig. 5, App. A.33) show that removing the global workspace yields the largest
drop in preference-evolution accuracy (≈14%), with further degradation when modules such as Emotion
or Memory are removed. We will add a small table of calls/tokens/latency per decision so it is
explicit that gains come from the GNWT architecture plus CogniPair, not simply more compute.

**Behavior, domains, and human study.** On the Columbia Speed Dating dataset, CogniPair builds 551
GNWT-Agents. The same architecture transfers to job-interview simulations, where candidate and
hiring-manager twins reach 81% decision concordance. Our human study finds realistic behavior, high
decision concordance in both domains, and strong agreement between our adventure-style assessment and
a BFI-style inventory; each participant evaluates one GNWT-based twin. Directly comparing GNWT vs
baseline twins with human raters is valuable future work and will be listed in the Limitations
section.

**Demographics.** We strictly follow the original Columbia Speed Dating encodings and will add a
short table summarizing the anonymized attributes used. These fields feed the Persona & Trait
Initialization module to construct P and stable priors; we do not re-bin or re-identify any
category.

**Positioning and privacy.** We will also clarify positioning and presentation. Our setting differs
from ToM sandbox agents and workflow-optimizing systems for persistent worlds or code/math/QA; here
we study internal GNWT with personality adaptation on fixed, real-world social protocols (dating and
hiring), comparing against four standard agent families instantiated on the same human data under
matched compute. Our current architecture does not maintain explicit nested belief states;
ToM-like reasoning is distributed, and we will note that an explicit ToM module could be added as
another broadcaster in the workspace. Given the sensitivity of digital-twin dating and hiring, we
will highlight our privacy stack.

---

### Meta-Review · Area_Chair_mLv1 · 2026-01-05

**Summary:**

The paper presents GNWT-Agents, a cognitive architecture inspired by Global Workspace Theory for creating digital twins. Five modules (Emotion, Memory, Planning, Social Norms, Goal-Tracking) are coordinated via a global workspace broadcast mechanism. Evaluated on speed dating (551 agents from Columbia dataset) and job interviews, achieving 72% correlation with human attraction patterns and 72.5% preference evolution accuracy vs 61.3% for baselines. Human validation shows 5.6/7 behavioral accuracy rating.

**Reviewer Concerns:**

Addressed: Authors provided detailed GNWT loop formalization with pseudocode in rebuttal. Clarified that personality vector P is separate from Memory module. Added compute comparison showing GNWT uses 7 calls per decision vs 1-5 for baselines, with modules running in parallel. Ablations show removing global workspace causes 14% drop, supporting architectural contribution over compute advantage. Cross-domain generalization demonstrated via job interview results (81% concordance).

Outstanding: Several substantive concerns remain unresolved.

First, Theory of Mind. EBNQ (rating 4, conf 5) noted ToM is "inevitable when talking about dating tasks" but the architecture lacks explicit partner belief modeling. Authors acknowledge ToM reasoning is only "distributed" across modules with no dedicated belief states. For a paper claiming psychological authenticity in social interaction, this is a significant gap.

Second, the human validation methodology. HnWZ (rating 4, conf 4) correctly identified that the human study only evaluates GNWT agents, not baseline agents. Without head-to-head comparison, the 5.6/7 rating and 74% agreement cannot be attributed to the GNWT architecture specifically. Authors acknowledge this limitation but did not address it.

Third, technical clarity. Multiple reviewers (q5Zt, zfz3) found the method description vague even after reading the paper. The figures were called "too abstract" and "conceptual." While authors promise revisions, the current submission lacks sufficient technical detail for reproducibility.

Fourth, scientific grounding of module choice. q5Zt (rating 4, conf 2) questioned why these five modules and not others like attention or reasoning. Authors' answer that "attention is the salience step" and "reasoning is handled by the LLM" is somewhat circular - it does not justify why these specific cognitive functions were chosen over alternatives from the cognitive science literature.

**Reviewer Scores:**

E4Ms (8, Conf 4): Strong accept. Found the work well-written and comprehensive. Only positive reviewer.

EBNQ (4, Conf 5): Marginally below. High confidence, concerned about missing ToM, latency analysis, and SOTA social agent comparisons.

q5Zt (4, Conf 2): Marginally below. Low confidence but unconvinced by scientific support for module design.

zfz3 (4, Conf 4): Marginally below. Appreciated motivation but found technical details insufficient.

HnWZ (4, Conf 4): Marginally below. Questioned whether gains come from compute vs architecture, and flagged human study methodology.

Final distribution: 8/4/4/4/4. One strong accept, four marginally below. Average 4.8.

---

### Decision · Program_Chairs · 2026-01-26

Reject